# Immune pathways and *TP53* missense mutations are associated with longer survival in canine osteosarcoma

Sunetra Das [1,2 ✉], Rupa Idate[1,2], Daniel P. Regan[2,3,4], Jared S. Fowles[1,2,5], Susan E. Lana[1,2], Douglas H. Thamm[1,2,4,5], Daniel L. Gustafson[1,2,4,5] & Dawn L. Duval [1,2,4,5 ✉]

Osteosarcoma affects about 2.8% of dogs with cancer, with a one-year survival rate of approximately 45%. The purpose of this study was to characterize mutation and expression profiles of osteosarcoma and its association with outcome in dogs. The number of somatic variants identified across 26 samples ranged from 145 to 2,697 with top recurrent mutations observed in *TP53* and *SETD2*. Additionally, 47 cancer genes were identified with copy number variations. Missense *TP53* mutation status and low pre-treatment blood monocyte counts were associated with a longer disease-free interval (DFI). Patients with longer DFI also showed increased transcript levels of anti-tumor immune response genes. Although, T-cell and myeloid cell quantifications were not significantly associated with outcome; immune related genes, *PDL-1* and *CD160*, were correlated with T-cell abundance. Overall, the association of gene expression and mutation profiles to outcome provides insights into pathogenesis and therapeutic interventions in osteosarcoma patients.

[1] Department of Clinical Sciences, College of Veterinary Medicine and Biomedical Sciences, Colorado State University, Fort Collins, CO 80523, USA. [2] Flint Animal Cancer Center, Colorado State University, Fort Collins, CO 80523, USA. [3] Department of Microbiology, Immunology, & Pathology, Colorado State University, Fort Collins, CO 80523, USA. [4] University of Colorado Cancer Center, Anschutz Medical Campus, Aurora, CO 80045, USA. [5] Cell and Molecular Biology Graduate Program, Colorado State University, Fort Collins, CO 80523, USA. ✉email: sunetra.das@colostate.edu; Dawn.Duval@colostate.edu

Osteosarcoma (OSA) is a neoplasm of bone (mesodermal origin) that occurs spontaneously in a wide spectrum of mammals including humans and dogs[1]. OSAs commonly arise in the metaphysis of long bones in both dogs and humans, and produce an extracellular matrix called tumor osteoid. This is the most common type of bone cancer in children, adolescents, and to a lesser extent in the aging adult population (>60 years old). In comparison, 80% of OSA occurs in dogs >7 years of age and rarely in juveniles (6–8%)[2,3]. OSA risk has been defined as breed dependent with increased incidence in large and giant breed dogs[4]. According to the Surveillance, Epidemiology, and End Results (SEER; https://seer.cancer.gov/) database, the five-year survival rate in humans is 66% but is only 27% in patients with measurable metastatic disease at diagnosis. Current treatments in humans include limb sparing surgery and neoadjuvant or adjuvant chemotherapy using doxorubicin, cisplatin, methotrexate, and ifosfamide[5] increasing the five-year survival rate from 20 to 70%. Osteosarcoma in dogs is typically treated by amputation of the affected limb followed by chemotherapy with doxorubicin and/or platinum-based therapies. A 2014 study of 470 dogs treated for OSA with amputation and chemotherapy found that the median disease-free interval (DFI) was 291 days, and was not statistically different based on the type of chemotherapy[6].

The discovery and targeting of genomic modifications that lead to malignancies is possible due to advancements in sequencing technology and computational analysis. Recent articles have detailed the genomic drivers of OSA in both humans and dogs[7–12]. The key discoveries in human OSA are large numbers of structural and copy number variations, with few activating mutations in oncogenes and infrequent point mutations in protein coding genes. One of the first papers to conduct whole genome sequencing in human OSA identified *TP53* structural variants (SV) and single nucleotide variants (SNV) in 55 and 45% of their 20 samples, respectively[7]. Other tumor suppressor genes with recurrent somatic variants were *RB1*, *ATRX*, and *DLG2* in 29−53% of the patients. Perry et al.[8] reported similar variants and identified the PI3K/MTOR pathway, altered in 24% of patients, as a therapeutic target[8]. Whole genome and RNA-sequencing analysis in patient tumors and patient-derived xenografts have demonstrated that genes with somatic copy number alterations can be targeted to reduce tumor burden[10]. To date, there have been two large-scale reports of genome and exome wide variant analyses in canine OSA[11,12]. In addition to recurrent *TP53* point mutations and CNVs, these studies identified two other recurrently mutated genes: *SETD2* (histone lysine methyltransferase) and *DMD* (dystrophin) not previously identified in human OSA. However, it is not clear if these genes represent cancer drivers in dogs. Like human OSA, the short variant mutational burden was low in comparison to structural (SV) and copy number variants (CNV) in canine bone tumors.

Immunotherapy is emerging as an alternative treatment for many cancers. The tumor microenvironment profile and activation of macrophages and monocytes by the bacterial cell wall analog, L-MTP-PE, in canine and human OSA suggested that OSA might be receptive to immune therapies[13]. Recent articles on the immunogenomic landscape in human OSA have sought to identify prognostic markers and genomic targets for immune therapy[14–16]. Expression of *PD-L1* in human OSA was significantly associated with immune infiltrates such as T cells, dendritic cells, and natural killer cells[17]. However, low infiltrate levels could explain limited success in treating OSA patients with immunotherapy[16,18].

We have conducted multi-platform analysis of 26 canine OSA samples, including whole exome sequencing, microarray analysis, and immune cell profiling. Like previous published work, we report a prevalence of CNVs over short variants (SNVs and INDELs). The top two recurrently mutated cancer genes with short variants were *TP53* and *SETD2*. Using GISTIC2 to identify CNV, we found more gene deletions than amplifications. Additionally, we identified differentially expressed genes between tumors and normal metaphyseal bone based on Affymetrix Canine 2.0 microarrays. The observed variant and gene expression data were correlated with patient outcome data following treatment with limb amputation and doxorubicin and/or platinum-based therapies. The disease-free interval (DFI) was used to categorize the patients in short (DFI < 90 days) and long (DFI > 458 days) groups. Tumors from the long DFI patients were enriched for genes in immune-related pathways. In summary, the current work explores the relationship between the canine OSA mutational spectrum and associated changes in gene expression to identify pathways that contribute to cancer progression and therapeutic sensitivity.

## Results and discussion

**Variant quantification and mutational landscape in canine OSA.** The whole exome sequencing data from 26 primary OSA tumors and 26 matched normals were analyzed to identify somatic short variants, i.e., single nucleotide variants (SNVs) insertion and deletions (INDELs), and copy number variations (CNVs) (Supplementary Data 1). Microarrays were processed from 108 OSA samples as well as 8 normal bone samples (Supplementary Table 1 and Supplementary Data 2). Additionally, clinical outcome data was used to identify pathways and somatic variants that were modulated based on disease-free intervals (DFI) (Supplementary Data 2). The median depth of sequencing for normal and tumor samples was 247X (range: 88X–578X) and 295X (range: 126X–453X), respectively (Supplementary Fig. 1). The total number of somatic short variants identified across 26 primary tumor samples ranged from 145 (T-1247) to 2697 (T-153) (Fig. 1a). Of these variants, 6.9% (T-1272) to 25.9% (T-554) were protein coding variants. The protein-coding mutations per megabase ranged from 0.25 (T-1272) to 7.39 (T-153) (Supplementary Fig. 2). One sample, T-153, with a high mutational burden, could be considered hypermutable using criteria established by Gröbner et al.[19]. There was no significant correlation between DFI and mutations per megabase (Hazard Ratio: 1.095, $p = 0.5$). This suggests that absolute numbers of tumor mutations were not primary predictors of outcome in canine OSA. Within the protein coding short somatic variants, an average of 80% were missense mutations (range 64–92%) (Fig. 1b). Overall, there were 739 deleterious and 889 tolerated missense mutations as identified by SIFT scoring (Supplementary Data 3). A single metastatic tumor sample (M-1166) had a total of 746 somatic variants of which 10.9% were located within protein-coding regions of genes. In comparison, the corresponding primary tumor (T-1166) had 908 somatic short variants and 14.6% of these were located within coding regions.

The CNVs were analyzed using Sequenza and GISTIC to identify significantly amplified and deleted regions. Similar to human OSA, the samples in this study had more CNVs than SNVs and INDELs[7]. The median number of genes with significant CNVs was 1468 with a range from 749 (T-1247) to 1630 (T-153). The majority of CNVs were deletions which ranged from 81% (T-1247) to 90% (T-458) (Supplementary Fig. 3). The number of CNV genes in the metastatic tumor, M-1166, was 1356 (91.5% deletions). In comparison, the primary tumor, T-1166, had 1156 CNV genes (98.6% deletions). There was no significant association between DFI and total CNVs among the 26 samples (Hazard Ratio: 1.001, $p = 0.4$), suggesting that the number of genes with CNVs was not prognostic.

The distribution of six types of single nucleotide substitutions ($C \cdot G \rightarrow A \cdot T$, $C \cdot G \rightarrow G \cdot C$, $C \cdot G \rightarrow T \cdot A$, $T \cdot A \rightarrow A \cdot T$, $T \cdot A \rightarrow C \cdot G$, and $T \cdot A \rightarrow G \cdot C$) for all 26 samples revealed $C \cdot G \rightarrow T \cdot A$ transition mutations as the most frequent substitution (Supplementary Fig. 4A). This is similar to distributions reported in previous

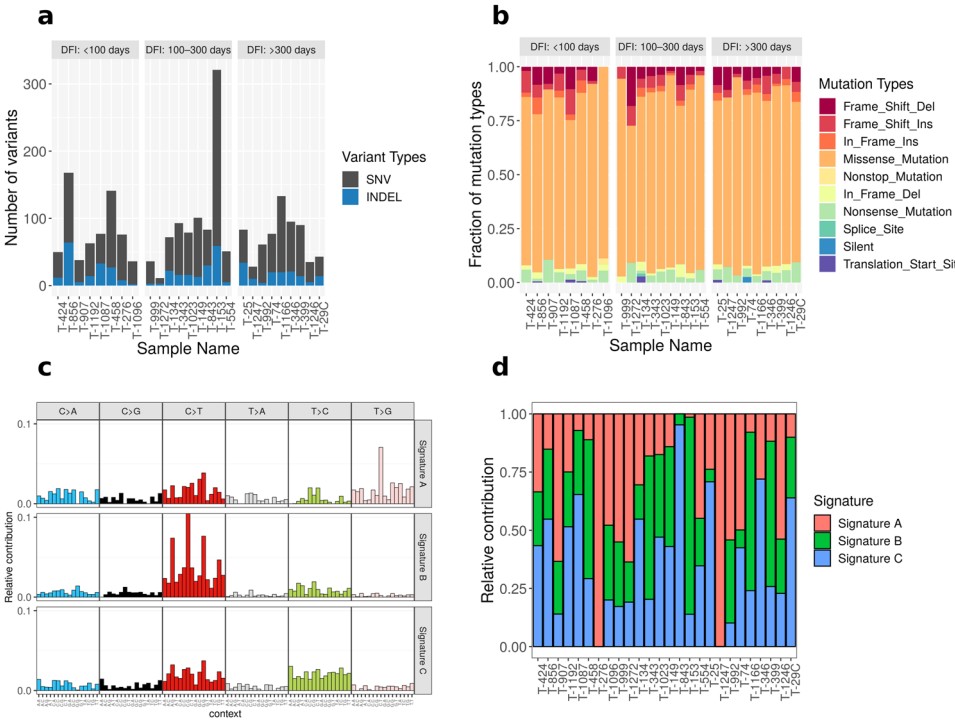

**Fig. 1 Mutational signature profile in canine osteosarcoma. a** Distribution of total short variants (SNVs and INDELs) across 26 samples. The samples were sorted in ascending order of disease-free interval (DFI) and binned in three discrete groups. **b** Distribution of 10 different mutation types as identified by Ensembl Variant Effect Predictor across 26 samples. Missense mutations were the most common type of short variant in this dataset. **c** Mutation signature analyses showing the relative distribution of six single nucleotide changes in 96 different trinucleotide contexts across three de novo signatures. The de novo signatures were extracted from frequency of single nucleotide changes in 26 tumors using non-negative matrix factorization (NMF) method. **d** Relative contribution of three predominant de novo signatures across 26 samples. Source data for these plots are included in the Supplementary Data 11.

WES and WGS studies of canine OSA[11,12]. Using the non-negative matrix factorization (NMF) approach three top-ranked de novo signatures (Signatures A, B, and C) were generated from the frequency of 96 types of trinucleotide context SNVs in 26 osteosarcoma samples (Fig. 1c). Signature A was similar to COSMIC signature 9[20] (cosine similarity = 0.72) which is characterized by mutations attributed to DNA polymerase η. DNA Polymerase η is recruited to sites lacking bases during the repair of U:G mismatches generated by activation-induced cytidine deaminase for the somatic hypermutation used to generate immunological diversity and is often found in chronic lymphocytic leukemias and malignant B-cell lymphomas[21]. Signature B was similar to COSMIC signature 1 (cosine similarity = 0.89) which is characterized by spontaneous deamination of 5-methylcytosine resulting in C → T transitions. This signature correlates with age of cancer onset in humans. Signature C was similar to COSMIC signature 5 which is characterized by transcriptional strand bias for T > C substitutions at ApTpN context and is found in most cancer types. Overall, 38, 24, and 38% of samples resembled A, B, and C, respectively (Fig. 1d). Additionally, we compared distribution of the trinucleotide context of 26 tumor samples to 30 known COSMIC signatures (v1). The majority of the OSA samples were most similar to either COSMIC signature 1 or 5; however, one sample had similarity to signature 9 (T-1247). There were four samples with highest cosine similarity for signature 6 (T-1192, T-907, T-992, T-999); and one each similar to signature 17 (T-276) and signature 19 (T-1272) (Supplementary Fig. 4B).

Sakthikumar et al. identified COSMIC signatures 1 and 17 and found a greater representation of signature 1, associated with aging, in Rottweilers and Greyhounds, while COSMIC signature 17 was more common in Golden Retrievers. Like the Gardner et al. study, we identified high similarity to COSMIC signature 17 in only one

sample (Supplementary Fig. 4B). This sample (T-276) was from a female mixed breed dog that fell within the DFI < 100-day group, had 1.91 protein coding mutations per megabase, and is *TP53* wildtype. Given the rarity of this signature in our samples, this sample was binned as signature A. While only identified in the WGS analysis, the Gardner et al. study also identified a group of tumors with COSMIC signature 9.

**Short somatic variants**. Of the somatic short variants, 6.9–25.9% were identified in protein coding genes. A total of 1579 protein coding genes had a variant in at least one of the 26 samples for a total of 1934 protein coding variants (Supplementary Data 3). With a range of 321 (T-153) to 11 (T-1272) genes with variants across 26 samples, only 14 genes were recurrently mutated in at least 15% of the samples (Supplementary Fig. 5). There were 1100 genes that were mutated in only one sample and 129 genes mutated in a maximum of two samples. Protein-coding genes with variants were binned to identify enriched annotation terms (Supplementary Data 4). Selected pathways associated with these genes included ECM interaction, Focal adhesion, cell cycle, PI3K-Akt-, and Calcium-signaling, like pathways bearing somatic variants previously identified in pediatric OSA[8].

The protein coding somatic variants were also filtered for known cancer genes using the curated dataset from the Cancer Gene Census[22]. On average across the 26 samples, cancer genes represented 5.9% (±3.3%) of coding mutations with 62 cancer genes identified. The top four genes (mutated in at least in 12% of samples) were *TP53*, *SETD2* (SET domain containing 2, histone lysine methyltransferase), *HSP90AA1* (heat shock protein 90 kDa alpha, member A1), and *DNMT3A* (DNA-methyltransferase 3A) (Fig. 2a). The variants identified in *TP53* were primarily located

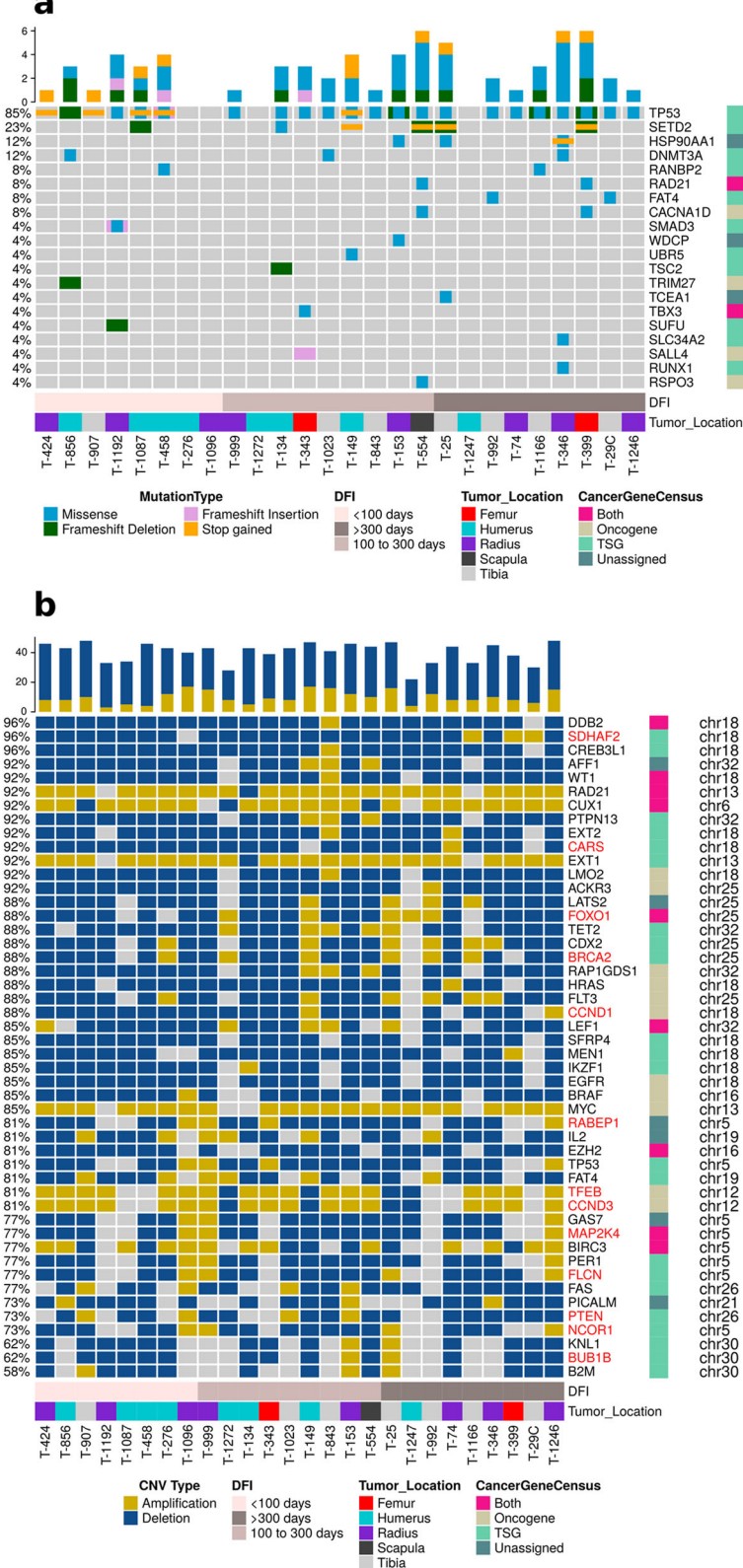

**Fig. 2 Oncoplot of cancer census genes with variants in canine osteosarcoma samples. a** The genes with short variants (SNVs and INDELs) and **b** with recurrent copy number variations (amplifications and deletions) were plotted along with disease-free interval (DFI) and tumor location. The cancer genes were categorized as oncogenes and tumor suppressor genes in accordance to the Cancer Gene Census. The samples were sorted in ascending order of DFI in both plots. Gene names colored red in panel **b** had a significant correlation between copy number amplitude and corresponding transcript expression.

within the DNA binding domain and are considered driver mutations in human cancers (Supplementary Fig. 6; Supplementary Data 5). Over 80% of the samples had at least one *TP53* short variant and/or CNV similar to previous reports in both human and canine studies[8,11,12]. In comparison, Chen et al. identified *TP53* pathway mutations in each of 20 human tumors examined[7]. This study, with only 2 somatic missense mutations, one frameshift variant, and 55% (11/20) bearing structural variants in the first intron, suggests that for human OSA, TP53 missense variants are comparatively rare.

The second most frequently mutated gene with short variants was *SETD2* (6 samples). This gene is a potential tumor suppressor and is mutated in several human solid tumors, including OSA[2,23]. Three samples had both frameshift and stop-gained mutations in SETD2 (T-544 and T-399: P1158Lfs, R396*; T-25: L124Yfs, Q1546*), three samples had either missense (S1658P in T-134), frameshift (Y1033Ifs inT-1087), or stop-gained mutations (Q1431* in T-149). Homologous frameshift and stop gained mutations of *SETD2* in human cancers are considered likely oncogenic. The missense mutation identified here in *SETD2* is not considered to be a cancer hotspot as reported by pan-cancer analysis in cBioPortal. Although the majority of the *SETD2* variants resulted in frameshifts and/or nonsense mutations, its status did not impact transcript expression based on Canine 2.0 Affymetrix microarray analysis ($p = 0.2$, Wilcoxon rank sum test between expression values of samples with mutation and wildtype for SETD2). While this suggests that *SETD2* variants fail to significantly impact transcript levels, the functional significance of these alterations is unknown currently.

The *HSP90AA1* gene was mutated in 12% of the samples and all three samples carried the same missense mutation (A149D in T-346, T-153, T-25). The only other recurrently mutated gene, *DNMT3A*, carried two different missense mutations in three samples (N597S in T-346 and T-856; W738R in T-1023). Of the 62 cancer genes, 58 genes were mutated in only one or two of the samples, limiting further analysis regarding their impact on clinical outcome and suggesting that they may be passenger mutations. In addition, there was limited overlap between the less frequently mutated genes in our study and the two other canine studies[11,12].

**Copy number variants**. In comparison to short variants, we identified more genes with somatic CNVs. A total of 1662 genes were significantly altered across 26 samples (Supplementary Data 5). The median number of genes with significant amplifications and deletions was 169 (range: 114: T-1166 to 205: T-346) and 1295 (range: 607: T-1247 to 1460: T-458), respectively. Functional characterization of these genes identified enriched KEGG (q-value <0.1) pathways including MAPK signaling, Autophagy, PI3K-AKT, p53 signaling, and FOXO signaling pathways (Supplementary Data 7). All of these signaling pathways are known to be altered in canine and human OSA[8,12]. A total of 47 cancer genes (COSMIC Cancer Gene Census, v91) had significantly deleted or amplified copy number aberrations (Fig. 2b). Six cancer genes were significantly amplified, including, *CCND3* (cyclin D3), *CUX1* (cut like homeobox 1), *EXT1* (exostosin glycosyltransferase 1), *MYC* (MYC proto-oncogene), *RAD21* (RAD21 cohesin complex component), and *TFEB* (transcription factor EB). Most of the remaining 41 genes were significantly deleted. *FOXO1* was both significantly deleted and amplified in individual samples.

Perry et al.[8] identified 3450 genes that had significant copy number alterations in pediatric OSA. We found 105 genes that overlapped, including *MYC*, *CCND3*, and *TFEB* (Supplementary Data 6). However, only 35 CNV genes were common between

this study and another canine whole exome sequencing study[11]. This could be due to variation across canine breeds for OSA samples and/or differences in the protocols used for sequencing and bioinformatics analysis.

The functional effect of copy number variations was evaluated through Pearson correlations with gene expression. Of the 1662 genes with recurrent copy number alterations, 1471 genes had gene expression values from microarray data for all 26 samples. There were 256 genes with significant Pearson correlation coefficients between CNV amplitude and expression (Supplementary Data 8). Of these, 13 were cancer genes which included *BRCA2*, *BUB1B*, CARS, *CCND1*, *CCND3*, *FLCN*, *FOXO1*, *MAP2K4*, *NCOR1*, *PTEN*, *RABEP1*, *SDHAF2*, and *TFEB* (Supplementary Fig. 7). Thus, while a plethora of CNV changes were identified, significant correlation with gene expression was limited to 15% of the identified CNV genes which supports their biological relevance. Notably, *MYC* was amplified in 85% of the current samples but was not significantly correlated with increased expression within the Canine 2.0 microarray data. In contrast, whole genome sequencing identified *MYC* CNVs in only 38% of canine and 39% of human osteosarcomas[10,12], while *MYC* amplification was associated with concomitant elevated transcript expression only in the canine study. Array CGH analysis has also identified recurrent *MYC* amplifications in canine OSA, confirmed by fluorescent in situ hybridization, but not associated with elevations in *MYC* transcript expression[24], suggesting that some instances of *MYC* gene amplification may not result in elevated gene expression. In addition, previous studies in human and canine OSA have identified *ATRX* and *DLG2* as SV or CNV[7,12]. Similarly, we identified *DLG2* copy number losses in 62% of our OSA samples; however, the CNV values did not significantly correlate with *DLG2* transcript expression ($R = 0.21$, $p = 0.3$). This may suggest that variability of within and across-species CNV comparisons might result from poor performance of WES CNV analyses algorithms[25].

**Over-representation of extracellular matrix components in canine osteosarcoma samples**. Using microarray analysis, we compared the transcriptome profiles of 26 canine OSA samples and eight normal bone samples. The number of up- and down-regulated probes (adjusted $p$-value <0.05 and $\log_2$ fold change > ±2) were 201 and 721, respectively (Supplementary Fig. 8), resulting in 585 differentially expressed genes (DEGs) (Fig. 3a). The DEGs were analyzed using pre-ranked Gene Set Enrichment Analysis (GSEA) to identify enriched pathways and GO terms in the canine OSA tumor samples. A total of 187 gene sets were enriched at an FDR of <0.05 (Supplementary Data 9). Multiple gene sets associated with extracellular matrix (ECM), cell proliferation, epidermal to mesenchymal transition, glycolysis, and metastasis genes were enriched (Fig. 3b–c, Supplementary Data 9) in the tumor samples when compared to normal bone (positive NES score). Several pathways pertaining to metastatic cancer were enriched in these primary tumor samples. The ECM components upregulated in tumors included integrins, collagen, proteoglycan, and glycoprotein genes. ECM dysregulation leads to progression of osteosarcoma and promotes metastasis[26]. Culturing an OSA cell line in the presence of ECM resulted in doxorubicin resistance and decreased *TP53* protein levels, suggesting a mechanism for drug resistance[27].

The gene sets downregulated in tumors (negative NES score) were: myogenesis, innate and humoral immune response, paracrine hedgehog signaling, and drug transport (Fig. 3d, Supplementary Data 9). A component of the myogenesis gene set, *Duchenne muscular dystrophy* or *dystrophin* (*DMD*), had major structural rearrangements resulting in deletion in about 50% of canine OSA

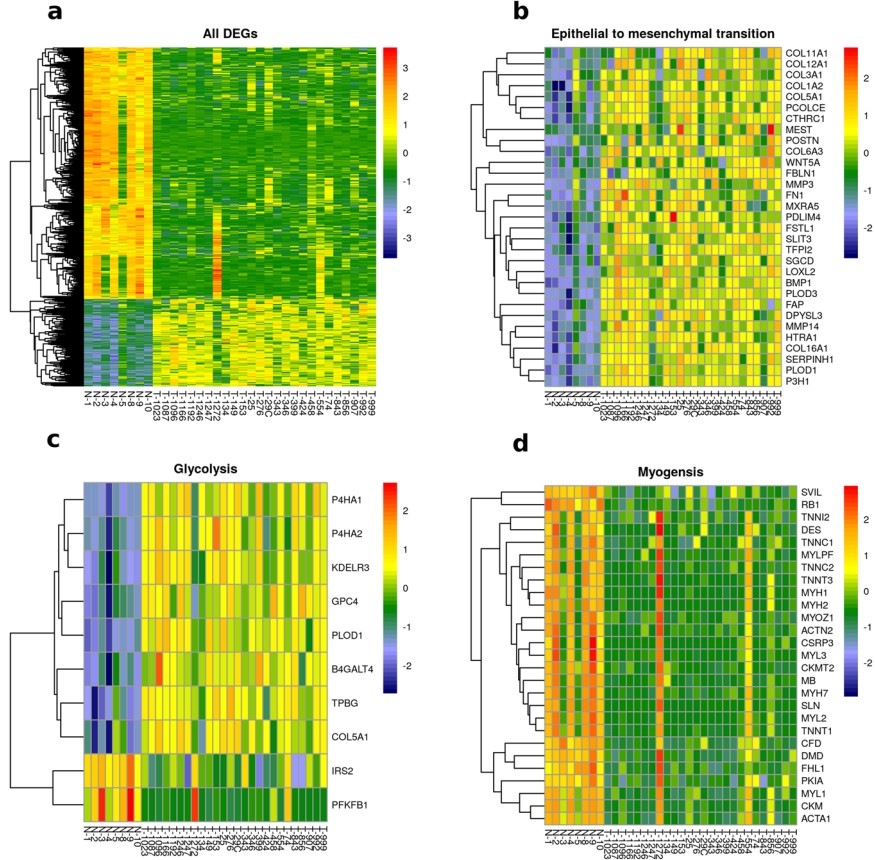

**Fig. 3 Graphical representation of differentially expressed genes (DEG).** **a** Heatmap of DEGs between normal bone (N-) and canine osteosarcoma samples (T-). **b–d** Heatmaps of core-enriched genes associated with three significantly enriched Hallmark gene sets (epithelial to mesenchymal transition, glycolysis and myogenesis), as identified by GSEA analysis. These genes were differentially expressed in tumor samples when compared to normal bone. The heatmap scale represents row z-score.

samples analyzed using whole genome sequencing[12]. Although these rearrangements could not be detected in our WES analysis, the average expression of tumor *DMD* was significantly lower than that in normal metaphyseal bone samples (Student's *t* test *p* value = 0.006). Loss of DMD in *mdx* mice is associated with reduced life span and these mice often develop rhabdomyosarcomas[28]. However, the effect of *DMD* deletion in OSA is still unknown. Consistent with a role for *RB1* loss in the development of osteosarcoma, the oncogenic signature gene set, RB_P107_DN.V1_UP, which includes genes with increased expression in response to the loss of RB1, was enriched in these canine tumors. *RB1* gene expression is down-regulated in the tumors relative to normal bone (log$_2$ fold change = −2.9, *q*-value = 5.9e-12), although copy number loss was not observed. In pediatric OSA development, deletion of *RB1* is observed in 29% of patients[7]. RNAseq analysis comparing the gene expression profiles between 4 normal and 3 OSA samples similarly identified dysregulated muscle and muscle contraction-related pathways as well as iron homeostasis and extracellular matrix genes[29]. Among the gene changes they confirmed using RT-qPCR and IHC analysis, we also saw elevated expression of *MMP3*, *SLC2A1*, *DKK3*, *POSTN*, and *ASPN* in OSA tumors compared to normal bone samples. A recent article reported loss of *PTEN* expression in canine OSA cell lines (POS/HMPOS) but continued expression of *RB1* in all four tested cell lines[30]. WES analysis of 8 canine OSA cell lines coupled with prior microarray analysis identified *TP53* missense mutations in 3 cell lines (Supplementary Fig. 14), and reduced *RB1* expression in 2 cell lines[31,32]. It also confirmed the loss of *PTEN* in HMPOS cell line, in addition to OSA8 and Abrams cell lines, and identified elevated *MDM2* expression in D17 cell line[31]. These gene

expression variations have also been reported in both canine and human OSA tumors.

**Immune response pathways were enriched in dogs with a long DFI.** For identification of pathways enriched in tumors from long DFI and short DFI patients, we used a larger microarray dataset of 108 tumor samples that included the 26 samples with WES data. We eliminated two of the 108 samples (T-474, T-1192) due to lack of follow up or death from other causes and sorted the remaining samples into two bins based on the 25th (short DFI, <90 days) and 75th (long DFI >458 days) percentiles of DFI. The transcriptome of these two groups (N = 27) was analyzed via GSEA to identify enriched pathways. Using the Hallmark and canonical pathways, 37 and 17 pathways/terms were found to be enriched in long and short DFI patients, respectively (Table 1 and Supplementary Data 10). The 37 gene sets upregulated in short DFI patients included cell cycle pathways, DNA repair pathways, and MYC and E2F targets. Heatmaps for selected genes sets KEGG DNA Replication and Biocarta ATR BRCA pathways, indicate elevated expression of genes from these pathways in the short DFI group (Supplementary Fig. 9). Activation of the ATR BRCA pathway might indicate that DNA repair pathways allow for cells proliferation in the face of replication stress. In addition, tumors from patients with a short DFI were enriched for validated targets of MYC activation suggesting that *MYC* may play an important role in this subset of aggressive tumors despite the lack of correlation between *MYC* expression and gene amplification. Similarly, enrichment of mTORC1 complex activated genes in short DFI patients substantiate its role in osteosarcoma

**Table 1 Summarization of gene sets and pathways that are enriched in short or long DFI patients.**

| Biological process | Enrichment category | Number of gene sets | Mean normalized enrichment score |
|---|---|---|---|
| Extracellular matrix | Long DFI patients | 8 | −1.97 |
| Immune system | Long DFI patients | 22 | −1.99 |
| DNA repair | Short DFI patients | 5 | 2.1 |
| Cell cycle | Short DFI patients | 5 | 1.9 |

See Supplementary Data 10 for details on enriched gene sets.

metastasis. The interplay of these factors with the observed enrichment of TP53 truncated and WT tumors in the short DFI group is a focus of future studies.

There were 22 immune-related and 8 ECM-related gene sets that were upregulated in long DFI patient tumors (Table 1, Supplementary Data 10). Heatmaps of Interferon alpha response and CD8 TCR pathway gene sets show upregulation in long DFI patients (Supplementary Fig. 10). It has been previously reported that higher immune infiltration is associated with better prognosis in both dogs and humans[33]. These authors identified an association between lower expression of immune-related genes, metastasis, and poor clinical outcome.

**Immune cell profiling.** Clinical trials over the past decade have demonstrated immune checkpoint inhibitors (ICI) to be an effective treatment modality in multiple human tumor types[34–38]. Despite substantial genomic instability in OS, clinical benefit of ICI therapy has not been observed in these patients, underscoring an immediate need for pre-clinical models that allow investigation of alternative therapeutic combinations that could shift this paradigm[7,39–41]. The negative results of ICI trials in OS are surprising, given the prior clinical success of the macrophage-activating innate immune stimulant Liposomal-Muramyl Tri-Peptide-PhosphatidylEthanolamine (L-MTP-PE) immunotherapy in this disease. L-MTP-PE was developed to stimulate antitumor activity of monocytes and macrophages, and has resulted in longer overall survival in both canine and human osteosarcoma patients in combination with chemotherapy[42,43]. Due to the extensive comparative similarities between canine and human OS[44,45], it is likely that clinical studies in dogs with OS hold potential to inform alternative combination approaches for human immunotherapy studies in this tumor type. To this end, we sought to characterize the immune landscape of canine OS as a prerequisite to informed immunotherapy trial design. Samples from the WES dataset with available tissue blocks were labeled with the pan-T cell marker CD3 and the myeloid cell marker MAC387 (S100A9) via immunohistochemistry (Fig. 4a–d). Immune cell density is reported as a percentage of total tumor area. The percent tumor area positive for CD3+ T cells ranged from 0.002% (T-25, DFI – 372 days) to 4.87% (T-29C, DFI – 1533 days as of 06/25/2021) (Fig. 4e). The percent tumor area positive for MAC387 + ranged from 0% (T-1023, DFI – 216 days) to 3.7% (T-74, DFI – 406 days) (Fig. 4f). Survival analysis using DFI showed no statistically significant difference in patients median stratified by levels of either CD3 or MAC387 staining (Supplementary Fig. 11).

We also used the gene expression data for more comprehensive immunogenomic profiling of the immune infiltrates in the tumor samples, given the lack of available canine-specific antibodies for immune phenotyping. This analysis was performed using 3 methods: gene expression profiling for immune cell types as previously described in Rooney et al., CIBERSORT, and the ESTIMATE R package[46–48] (Supplementary Table 2, Supplementary Fig. 12). Using Pearson correlations, we created a matrix to identify significant associations (FDR <0.05) between immune cell scores (derived from gene expression), mutations per megabase and quantified immunostaining of T cells (IHC T-cells) and macrophages (IHC-Macs). Positive correlations were observed between IHC T-cells and gene expression scores for cytolytic activity, co-stimulation T-cell, co-inhibition T-cell, CD8+ T-cell, MHC class I, and NK cells. Conversely, IHC-Macs staining did not correlate with the gene expression score for macrophages (Fig. 5a). This may not be surprising since MAC387 is ubiquitously expressed by all myeloid cells in the dog, including neutrophils[49]. However, the IHC-Macs score was positively correlated with co-inhibition T cells and co-stimulation APCs. Interestingly, the macrophage expression score was inversely correlated with CD4+ regulatory T cells, but positively correlated with neutrophils and MHC Class I, supporting a role for regulatory T cells in the suppression of these cell types (Fig. 5a).

Additionally, 85 genes used for generating immune signatures were individually correlated with IHC staining for T-cell and macrophage infiltration in tumors (Table 2). There were 18 genes that positively correlated with T-cell infiltration at a false discovery rate of <0.05, including the known co-inhibitory immune checkpoint molecules PDL1 (CD274), CD160 (ligand for Herpesvirus entry mediator), as well as the cytotoxic T-cell co-receptor CD8A (cell surface marker for cytotoxic T lymphocytes) (Table 2A). Only one gene, TNFSF4, was negatively correlated with T-cell infiltration, supporting a role for OX40 ligand in the proliferation and activation of cytotoxic T cells in mouse models of OSA[50]. Five genes, including PRF1, SLAMF1, OX40, TNFRSF9, and CXCR3 were positively correlated with MAC387 staining (Table 2B). The cytolytic activity (derived as geometric mean of perforin and granzyme A transcript expression) in OS tumors was significantly correlated with 21 immune response genes that are markers for T cells, macrophages, MHC Class I, dendritic cells, Type I and II IFN Response, B cells, and natural killer cells (Supplementary Table 3).

To further delineate the immune cell profiling with a larger gene dataset, we used the CIBERSORT deconvolution tool[47,51]. This analysis indicates a prevalence of both resting/uncommitted M0 and alternatively activated M2-polarized macrophages in all the tumor samples and lower numbers of M1-polarized pro-inflammatory macrophages (Fig. 5b). However, there was no correlation between the abundance of these three types of macrophages and clinical outcome. A recent CIBERSORT classification of human OSA similarly found that M0 macrophages dominated the immune landscape, and the group with the highest M0 representation had the worst survival probability and lowest ESTIMATE scores[52]. Infiltrating macrophages in studies of human (CD14+, CD68+, and CD163+) and canine OSA (CD204+) have correlated positively with outcome and their dominance of the immune cell composition of OSA may account for the positive clinical results with the macrophage activator, L-MTP-PE[52,53]. Immunohistochemical staining for CD3+ T cells positively correlated with the CIBERSORT score for four different cell types including: CD8 T cells ($R^2$: 0.91), activated mast cells ($R^2$: 0.98), plasma cells ($R^2$: 0.77), and gamma delta T cells ($R^2$: 0.73). Significant correlation of MAC387 positive cells with CIBERSORT immune scores was limited to follicular helper T cells ($R^2$: 0.67) indicating that in OSA the S100A9 marker may not be specific to myeloid cell lineages. A third method used to determine levels of immune infiltrates was the ESTIMATE algorithm[48]. The ESTIMATE immune score ranged from −83.61

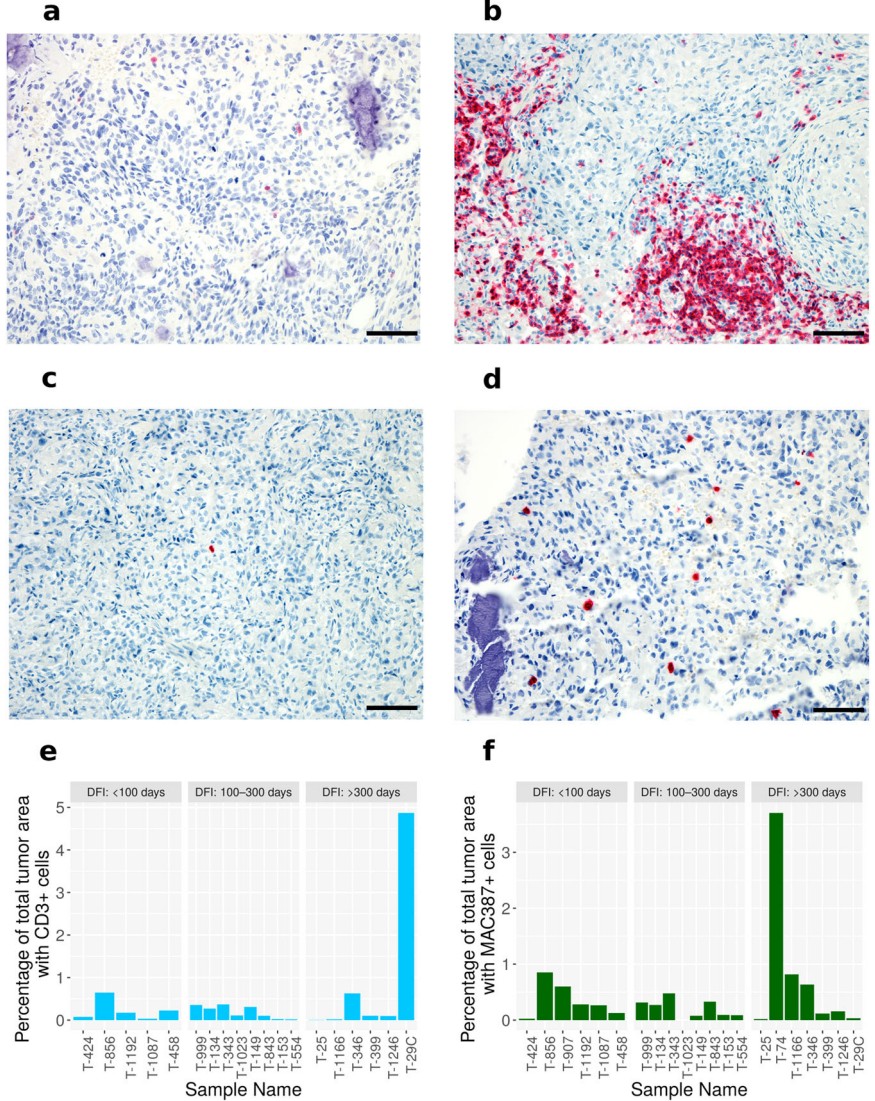

**Fig. 4 Immunohistochemical analysis of immune infiltrates in canine osteosarcoma samples. a, b** Representative photomicrographs of tumors demonstrating a low level (**a**; T-1023) and high level (**b**; T-1162) of CD3-positive immunolabeling of tumor-infiltrating T cells. 20x magnification. Fast red chromogen. Scale bars = 50 μM. **c, d** Representative photomicrographs of tumors demonstrating a low level (**c**; T-1162) and high level (**d**; T-856) of MAC387-positive immunolabeling of tumor-infiltrating myeloid cells. 20x magnification. Fast red chromogen. Scale bars = 50 μM. **e, f** Quantification of immunohistochemical labeling of T-cell (CD3+ cells) and Macrophages (MAC387 cells) across 19 and 22 samples, respectively. The samples were sorted in ascending order of disease-free interval (DFI) and binned in three discrete groups. Source data for plots **e** and **f** are included in the Supplementary Data 11.

(T-1087) 2110.4 (T-74) (Fig. 5c). ESTIMATE scores significantly correlated with IHC staining for both T cells (Pearson $R^2$—0.58, $p$ value—0.008) and macrophages (Pearson $R^2$—0.43, $p$ value—0.04). However, the ESTIMATE scores did not correlate with mutational burden or number of deleted and amplified genes. Additionally, there was no association between ESTIMATE immune score and DFI.

Thus, we have identified several immune response-related gene sets, primarily associated with an effector\cytotoxic T-cell response, which are upregulated or enriched in patients with long DFI. Similarly, a cross-species (human, dog, and mouse) transcriptome comparison found that gene cluster profiles for cell cycle and two immune signatures were commonly modulated across the three species in osteosarcoma tumors and cell lines[33]. However, there was no correlation between mutational burden and T-cell density and cytolytic activity in this study. These findings are consistent with a recently published study on the

immune landscape of OSA human patients[16], and suggest that, in contrast to other tumor types, the degree of mutational burden is unlikely to be an indicator for the presence of pre-existing anti-tumor immunity or immune therapy response in OSA.

Importantly, the significant positive correlation between transcriptomic CD8 T-cell and cytolytic activity with CD3 immunolabeling suggests that when present, T-cell infiltrates in canine OSA tumors are primarily of an effector CD8 cytotoxic T-cell phenotype, similar to their human counterparts. These data also demonstrate that while comprehensive immunophenotyping reagents for studies in dogs may be limited, CD3 IHC is a feasible and cost-effective surrogate for baseline anti-tumor immune response assessment in dogs. In this regard, we observed significant associations between CD3 T-cell infiltration and co-inhibitory immune checkpoint expression, including PD-L1, suggesting that when immune responses are present, similar mechanisms of T-cell immune exhaustion and adaptive immune

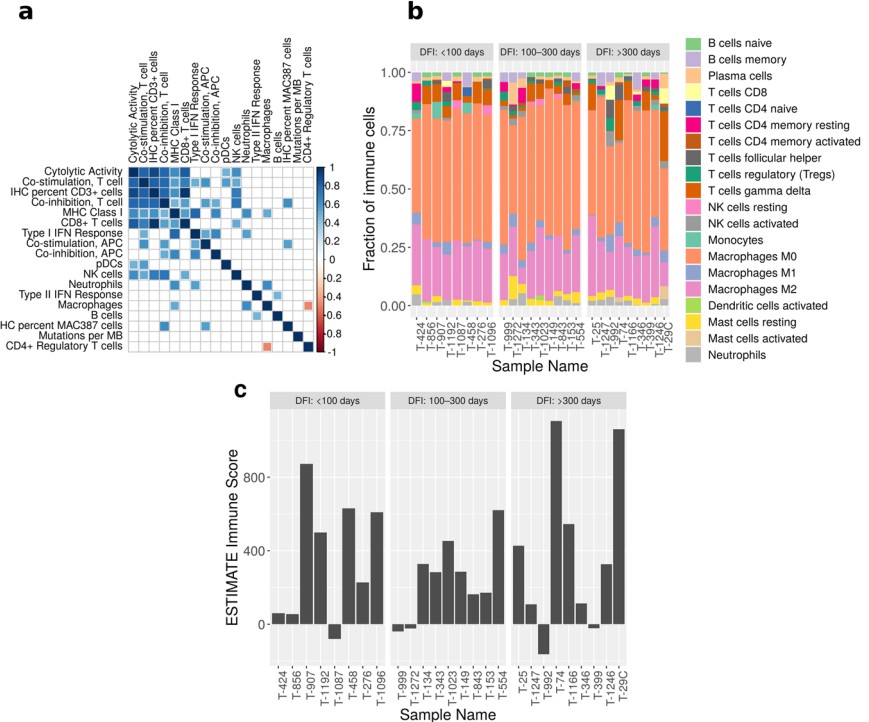

**Fig. 5 Immunogenomic profiling. a** Pearson correlation matrix of immune cell expression score and immune staining levels of T-cell (IHC percent CD3$^+$ cells) and macrophages (IHC percent MAC387 cells). We also correlated mutations per megabases (MB) with other variables. Only the significant (FDR <0.05) correlation values are plotted. The scale gradient color scale represents Pearson correlation coefficient. **b** Quantification of 20 immune cell types using CIBERSORT. The samples are sorted by DFI. The most prevalent cell type in the tumor microenvironment were M0 and M2 macrophages. **c** Immune score quantification using ESTIMATE R package. Source data for plots **b** and **c** are included in the Supplementary Data 11.

resistance are at play between human and canine tumors. While Wu et al.[16] reported that the ESTIMATE immune score is negatively correlated with the total number of deleted genes in human OS patients, we did not observe this association. This could be due to the method for identification of deleted genes in our study (whole exome sequencing) compared to the human OSA study (whole genome sequencing). Nonetheless, our gene set enrichment analysis and immunogenomic profiling suggests the presence of distinct subsets of so-called immunologically cold and hot tumors in canine OSA, and utilization of similar mechanisms of adaptive immune resistance in these tumors. These results provide an important foundation and rationale for designing immunotherapy studies in dogs as a translational strategy to improve solid tumor immunotherapy.

**Association of TP53 mutation status with clinical outcome**. To determine if the most frequently identified somatic variants had an impact on clinical outcome, we binned the tumors based upon the *TP53* mutation with the highest allelic frequency. Separated this way, we identified missense *TP53* mutations in 65% of the tumors, with frameshift or stop-gained (null) mutations identified in 23%, and wildtype (WT) *TP53* in 15% of tumors (Supplementary Data 5). We observed that missense mutations in *TP53* were associated with a longer DFI than either the *TP53* WT or null tumors following treatment by amputation of the affected limb and chemotherapy with doxorubicin and/or carboplatin (Fig. 6a). The median DFIs for patients with mutant and WT/ NULL *TP53* were 296 and 95 days, respectively [HR (95% CI) = 0.21 (0.08 to 0.58), p = 0.002]. The association of missense mutations in TP53 with a longer DFI was surprising since a previous study found no significant difference in DFI between dogs with tumors bearing *TP53* mutant and wildtype variants[54], but did identify a longer overall survival in dogs with wildtype

*TP53*. However, that study reported *TP53* variants in only 40.7% of cases, grouped missense mutations and frameshift or nonsense mutations for DFI analysis, and not all the dogs received chemotherapy. In contrast, each of the dogs in this study received at least 1 dose of chemotherapy. In addition, survival times can be confounded by euthanasia in companion animals, making DFI a potentially more biologically relevant measure of outcome, although some dogs may survive for long periods following the identification of metastasis. Using overall survival as the time event in Kaplan–Meyer analysis, dogs with *TP53* missense mutations continued to have statistically better outcomes than the wildtype/truncated group (Supplementary Fig. 13). A study in human small-cell lung cancer associated TP53 mutations identified in 54% of patients with longer relapse free intervals compared to patients with wild-type *TP53*[55]. Similarly, *TP53* mutant human cancers including breast, are significantly more likely to achieve pathological complete responses to chemotherapy[56–60]. Tumors in mice with the murine Tp53 R172H variant exhibit greater sensitivity to doxorubicin, and fail to exit the cell cycle following treatment, resulting in aberrant mitosis, and cell death[61]. Similarly, we found that among a panel of canine OSA cell lines, the HMPOS cell line which bears the homologous R to H variant[31], also had the greatest sensitivity to doxorubicin (Supplementary Fig. 14). Human *TP53* mutants, including R175H, R248W, and R273H, inactivate the ATM-dependent DNA damage response leading to chromosomal translocations and a defective G2/M checkpoint, and improved treatment responses[62]. A similar phenomenon may occur in canine OSA resulting in a longer DFI following chemotherapy in dogs bearing missense mutations in TP53. Conversely, forced expression of WT-p53 in p53 null A549 human lung carcinoma cells promoted cell survival in response to double strand break-inducing agents like doxorubicin by promoting cell cycle arrest and DNA damage repair[63]. We grouped

tumors with WT p53 with those bearing TP53 truncations to increase the numbers of tumors that lacked missense mutations. However, these *TP53* truncations (I247\*, R248\*, R265\*), homologous to human exon 6 truncations I195\*, hR196\*, and hR213\*, have the shortest DFIs among the WES group (Supplementary Fig. 15). In human cancers, exon 6 truncations have

no nuclear transcription regulatory activity; instead they are localized to the mitochondria and interact with cyclophilin to regulate the mitochondrial permeability transition pore. Exon 6 truncated TP53 increases reactive oxygen generation, epithelial to mesenchymal transition, and drives lung metastasis in melanoma cells[64,65]. Interestingly, there are 8 variants localized to AA245-265, including 5 additional missense mutations (Supplementary Fig. 6).

Along with *TP53* mutation status, we also evaluated six clinical co-variates that included age at diagnosis, tumor location (humerus versus other locations), sex, pre-treatment peripheral blood monocyte count, serum alkaline phosphatase levels, and body weight to identify their association with DFI via univariate Cox proportional hazards (COXPH) regression analysis. Two additional variables used in this analysis were tumor immuno-histochemical staining scores for CD3+ T cells and MAC387+ myeloid cells. Significantly increased risk was associated with two of the nine covariates, *TP53* mutation status and tumor location (Fig. 6a–b, Table 3). Although DFI among these 26 canine OS patients was not significantly lower in patients with increased numbers of peripheral blood monocytes in Kaplan–Meyer analysis (Fig. 6b), using five co-variates, *TP53* mutation status, tumor location, peripheral monocyte count, and alkaline phosphatase level (univariate COXPH $p < 0.2$), the final model for forward stepwise COXPH regression analysis indicated that wildtype/Null *TP53* mutation status and elevated peripheral blood monocyte counts were predictive of a shorter DFI (Table 4).

Thus, both high numbers of pre-treatment blood monocytes and wild type/null *TP53* may be markers of a poor prognosis in canine OSA treated by amputation and doxorubicin and/or carboplatin. Higher pre-treatment monocyte counts were previously reported to be associated with a shorter DFI in OSA patients[66]. Ghosh et al.[67] reported that mutant p53 reduces the activity of the cytoplasmic DNA sensing cascade which upregulates IFNB1 to stimulate CD8+, CD4+, and NK cells, while suppressing M2-tumor associated macrophages[67]. However, we found no difference in IFNB1 expression between tumors bearing mutant, wildtype, or truncated TP53 (ANOVA p-value: 0.435). Other studies have shown that mutant p53 can

**Table 2 Correlation of immune gene expression and immunohistochemical staining of T-cell and macrophage infiltration in the tumors.**

**A Gene expression correlation with CD3+ T cells in tumors.**

| Gene name | Associated immune cell type | Correlation coefficient | p-value | q-value |
|---|---|---|---|---|
| CD8A | CD8+ T cells | 0.94 | 1.4e-09 | 3.3e-07 |
| CD160 | Co-inhibition, T cell | 0.86 | 2.9e-06 | 0.0002 |
| CD274 or PDL1 | Co-inhibition, APC; Co-inhibition, T cell | 0.86 | 1.9e-06 | 0.0001 |
| GZMA | Cytolytic activity | 0.75 | 0.0002 | 0.003 |
| CD2 | Co-stimulation, T cell | 0.7 | 0.001 | 0.010 |
| ICOS | Co-stimulation, T cell | 0.68 | 0.002 | 0.015 |
| CLEC5A | Macrophages | 0.67 | 0.002 | 0.016 |
| TNFRSF4 | Co-stimulation, T cell | 0.65 | 0.003 | 0.020 |
| KLRF1 | NK cells | 0.65 | 0.003 | 0.020 |
| AHR | Type II IFN response | 0.64 | 0.003 | 0.021 |
| ISG20 | Type I IFN response | 0.63 | 0.004 | 0.022 |
| HAVCR2 | Co-inhibition, T cell | 0.63 | 0.004 | 0.023 |
| IRF8 | pDCs | 0.6 | 0.007 | 0.031 |
| CXCR3 | pDCs | 0.59 | 0.008 | 0.035 |
| TAP1 | MHC Class I | 0.58 | 0.009 | 0.036 |
| CD79B | B cells | 0.58 | 0.010 | 0.039 |
| KDM6B | Neutrophils | 0.56 | 0.012 | 0.046 |
| TNFSF4 | Co-stimulation, APC | −0.61 | 0.006 | 0.028 |

**B Gene expression correlation with MAC387+ myeloid cells in tumors.**

| Gene name | Associated immune cell type | Correlation coefficient | p-value | q-value |
|---|---|---|---|---|
| PRF1 | Cytolytic activity | 0.74 | 0.0001 | 0.008 |
| SLAMF1 | Co-stimulation, APC; Co-stimulation, T cell | 0.71 | 0.0003 | 0.011 |
| CD40 or OX40 | Co-stimulation, APC | 0.68 | 0.001 | 0.020 |
| TNFRSF9 | Co-stimulation, T cell | 0.65 | 0.002 | 0.028 |
| CXCR3 | pDCs | 0.62 | 0.003 | 0.042 |

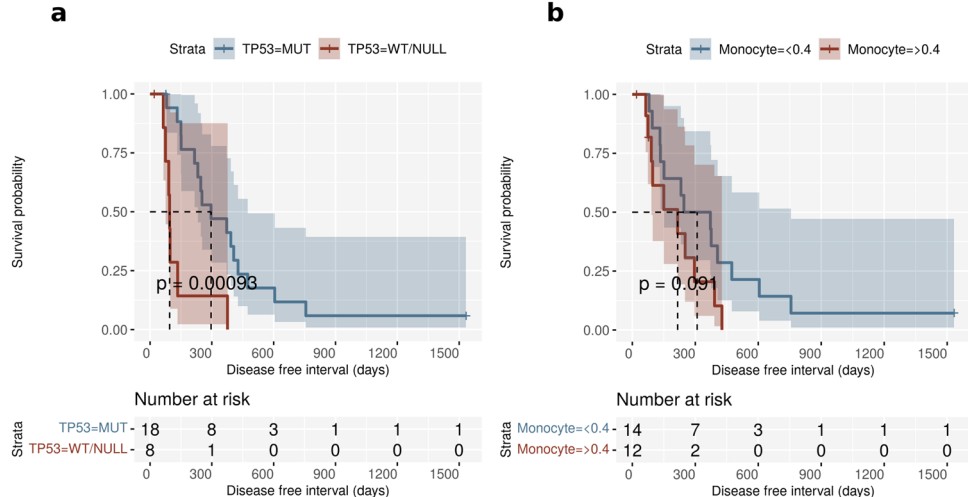

**Fig. 6 Association of TP53 mutation status and clinical parameters with outcome. a** Kaplan Meier plots for disease-free interval (DFI) in dogs with osteosarcoma stratified by *TP53* mutation status. The dogs with wild-type/null *TP53* status had a shorter DFI compared to dogs with *TP53* missense mutations. **b** Kaplan Meier plots for dogs with osteosarcoma stratified by pre-treatment monocyte count (> or <0.4 × 10³ cells/μl). Higher monocyte counts were associated with short DFI patients. The dotted lines in both plots represent median DFI for each stratum. Source data for plots **e** and **f** are included in the Supplementary Data 11.

**Table 3 Univariate Cox Proportional Hazard Modelling.**

| Variable | Group | Median survival time | Percent survival—1 year | Percent survival—2 years | HR | 95% CI | p value |
|---|---|---|---|---|---|---|---|
| TP53 mutation status | Mut | 296 | 47 | 12 | 0.21 | 0.08 to 0.58 | 0.002 |
| | WT/NULL | 95 | 14 | 0 | | | |
| Tumor location—Humerus | Yes | 114 | 12.5 | 0 | 4.05 | 1.51 to 10.84 | 0.005 |
| | No | 372 | 50 | 13 | | | |
| Pre-treatment peripheral blood monocytes (>or< $0.4 \times 10^3$ cells/µl) | High | 216 | 20 | 0 | 2.12 | 0.87 to 8.14 | 0.091 |
| | Low | 309 | 50 | 14 | | | |
| Alkaline phosphatase levels (>or< 140 units/liter) | High | 216 | 29 | 0 | 1.93 | 0.74 to 4.99 | 0.176 |
| | Normal | 246 | 35 | 6 | | | |
| IHC CD3 staining | NA | NA | NA | NA | 0.56 | 0.23 to 1.33 | 0.191 |
| Age | NA | NA | NA | NA | 1.13 | 0.93 to 1.36 | 0.224 |
| Sex | Male | 296 | 42 | 8 | 0.75 | 0.33 to 1.72 | 0.504 |
| | Female | 142 | 33 | 8 | | | |
| Weight | NA | NA | NA | NA | 1 | 0.97 to 1.04 | 0.846 |
| IHC MAC387 staining | NA | NA | NA | NA | 1.02 | 0.61 to 1.69 | 0.954 |

Results from univariate COXPH regression analysis using six clinical and one genomic parameter. The factors with a p-value <0.2 were used for the stepwise multivariate model.

**Table 4 Stepwise Cox Proportional Hazard Modelling.**

| Factors selected in stepwise model | HR | 95% CI | p value |
|---|---|---|---|
| TP53 mutation status (WT/NULL) | 6.39 | 2.12 to 19.209 | 0.0009 |
| Monocyte count (>$0.4 \times 10^3$ cells/µl) | 2.83 | 1.09 to 7.33 | 0.032 |

Stepwise COXPH multivariate analysis using four parameters for subset selection. The two significant co-variates reported here were associated with poor prognosis.

interact with NFκB to stimulate expression of genes involved in inflammation[68]. Further, interactions within the tumor microenvironment that impact the immune response may exhibit oncogene and tissue specificity[69–71]. Taken together, these studies suggest that oncogenic drivers, tissue of origin, and tumor microenvironment may all contribute to regulation of immune signaling.

## Conclusions

This study analyzed the genomic and transcriptomic profiles of 26 tumors from canine OSA patients and associated them with outcome data. Like other canine OSA studies, we have identified TP53 and SETD2 as the most recurrently mutated genes. In dogs treated with doxorubicin and/or carboplatin, wild-type/NULL TP53 mutation status was associated with short DFI. Additionally, higher expression of immune response genes was associated with long DFI patients. The immune-genomic profiling of tumors showed association of immune checkpoint genes and T-cell infiltration, but no association with genomic criteria such as mutational burden. Overall, this study indicates that the immune environment and TP53 mutation status are primary factors contributing to clinical response in canine OSA and consideration of these parameters may guide the development of alternative therapeutic regimens in human OSA.

## Methods

**Sample processing—bone tumors and matched normals**. Tumors from previously untreated dogs with OSA were collected along with blood, peripheral blood mononuclear cells (PBMC) or stroma as matched normal samples (Supplementary Data 1). The samples were flash frozen and kept at −80° C until processed for whole exome sequencing and microarray analysis. Following the manufacturer's protocol for TRIzol (Invitrogen), genomic DNA was extracted from 26 primary tumors, one metastatic tumor, and 26 matched canine blood, PBMC or stromal samples, and DNA was cleaned using DNeasy or QiaAMP DNA Blood mini kits (Qiagen). RNA was extracted from the same 26 tumor samples and 8 normal canine bone samples using TRIzol (Invitrogen) followed by RNeasy cleanup (Qiagen) for microarray analysis. Both RNA and DNA were quantified on a NanoDrop Microvolume Spectrophotometer and quality was assessed by TapeStation or Bioanalyzer (Agilent).

The whole exome DNA library was created and genomic exonic regions were captured using the Agilent SureSelect XT All Exon Canine V2 (part number: 931198, Santa Clara, CA) capture kit. This capture encompasses 43.45 Mb of canine exonic regions. The SureSelectXT Target Enrichment System for Illumina Paired-End Multiplexed Sequencing Library kit was used to create the genomic DNA library. The resultant library was sequenced on an Illumina HiSEQ4000 sequencer generating 151 bp paired end reads. RNA samples with a RIN value >8 were analyzed on GeneChip Canine 2.0 Genome Arrays (Affymetrix).

**Sample selection for retrospective study**. Primary tumors were selected from dogs meeting the following criteria: diagnosed with osteosarcoma and receiving treatment consisting of limb amputation and chemotherapeutic treatment protocols including at least one dose of platinum-based or doxorubicin-based chemotherapy or both. For inclusion in this retrospective cohort study, dogs must have had pre-operative thoracic radiographs or computed tomography and histopathologic confirmation of the diagnosis. Exclusion criteria were presence or suspicion of metastases at any site before amputation or prior treatment of appendicular OSA with radiation (palliative or curative-intent) protocols, chemotherapy, or surgery.

In most cases patients were monitored for disease progression and metastasis with physical exam and chest radiographs at 3-month intervals or sooner if clinical symptoms prompted additional screening. Pulmonary lesions were considered metastasis if they were multiple, progressive on serial radiographs, or confirmed on necropsy. Non pulmonary lesions were considered metastatic if they were suspicious for neoplasia based on radiographic or CT appearance or confirmed with histopathology either at the time of surgical removal or based on necropsy.

**Chemotherapy specifics**. Chemotherapeutic treatment of the 108 patients following amputation is detailed in Supplementary Data 2. Carboplatin as a single agent was administered using a protocol of 300 mg/m$^2$ given every 21 days for 4 or 6 treatments. Doxorubicin as a single agent was administered at 30 mg/m$^2$ every 21 days for 5 treatments. Alternating carboplatin and doxorubicin using the doses listed above, occurred every 21 days for a total of 6 treatments, 3 of each drug. Three dogs included in the microarray analysis received subcutaneous injections of a slow-release cisplatin polymer mixture (Atrigel) at a dose of approximately 60 mg/m$^2$.

**Mapping of whole exome sequence reads with BWA**. The Illumina reads were processed to identify somatic single nucleotide variants (SNVs), insertions and deletions (INDELs), and copy number variations (CNVs). The 150 bp reads were trimmed to eliminate low-quality (phred score <20) and adapter sequences by using the Trimmomatic tool (v0.36)[72]. Both the raw and trimmed/high-quality reads were assessed by FastQC (v0.11.5)[73]. The reads were then mapped against the canine genome (CanFam3.1) using the BWA (v0.7.15-r1140) tool[74]. The binary alignment files (BAM) were processed to mark duplicates and recalibrate bases prior to variant calling as per GATK (v4.1.2.0) best-practices[75].

**Indel and SNV calling using Mutect2 and post-processing of somatic variants**. The short somatic variants, SNVs and INDELs, were called from the BAM files using Mutect2. In addition to using a matched normal sample for each tumor, we created a panel of normals using 43 in-house samples (Supplementary Data 1). We also used 90 million population variants that were called from 722 dogs as the germline resource option within Mutect2 for calling somatic variants[76]. The variants were filtered using filterMutectCalls and variants with a PASS notation in the FILTER column were characterized as somatic variants. The variants were annotated using Ensembl Variant Effect Predictor (VEP, Ensembl version 99) and the VCF file was converted to MAF (Mutation annotation file) format using the perl code: vcf2maf.pl (https://github.com/mskcc/vcf2maf).

The mutational signature of each sample was deduced using the MutationalPatterns R package (v1.12.0)[77]. Using the mut_mat function the count matrix of 96 trinucleotide changes was derived from sample VCF files. To estimate the factorization rank, the non-negative matrix factorization (NMF) algorithm (NMF R package) was used. The de novo mutational signature was derived using the extract_signature function within the MutationalPatterns package. This resulted in decomposition of the mutation count matrix into three top ranked signatures and estimated the relative contribution within each sample. The sample signatures were compared to 30 known COSMIC signatures by calculating the cosine similarity between the NMF signatures and known cancer signatures.

The somatic variants were also processed to bin protein coding and cancer gene variants. The gene variants with HGVS.p variable notations were selected as protein coding genes. From this list, COSMIC (version 91) was used to identify variants within known cancer genes[22]. The oncoplots were plotted using R package ComplexHeatmap (v2.2.0). Pathway analysis of the mutated genes was conducted using DAVID Functional Annotation Bioinformatics Microarray Analysis (v6.8) tool and Enrichr to functionally annotate genes against Gene Ontology and KEGG databases[78,79].

**Copy number variant calling using Sequenza**. The allele-specific copy number variants were assessed using the Sequenza (v2.1.9999b1) tool[80]. Briefly, the sequenza-utils bam2seqz function was used along with paired BAM files (tumor and normal) to extract loci with A (major) and B (minor) allele frequencies. The sequenza R package used the output of bam2seqz function for GC content normalization of tumor normal depth ratio, allele-specific segmentation using the copynumber R package, model fitting to infer cellularity and ploidy parameters, and copy number profiles of tumors. The resulting segmentation file was annotated using the default options in the GISTIC2.0 tool to identify genes in the recurrently copy number altered regions of the tumor genome. A q-value cut-off of 0.01 was used to generate the final list of amplification and deletion peaks. The genes associated with these peaks were identified and cancer genes (COSMIC v91) were selected for plotting and cross-species comparison.

**Microarray analysis using RMA and limma**. Following the manufacturer's standard protocol, RNA from 26 tumor and 8 normal bone samples were transcribed to cDNA, labeled, and hybridized onto GeneChip™ Canine Genome 2.0 Array (ThermoFisher Scientific, catalog number: 900727), and scanned on an Affymetrix Scanner 3000. The data files (CEL) were processed in R for normalization of probes and differential expression of genes. Using simpleaffy R package (v2.62.0), the CEL files for both normal and tumor samples were normalized by Robust Multi-Array Average (RMA) method, and batch corrected by ComBat function which generated $\log_2$ expression values for 43,035 probes. The differential expression analyses between normal and tumor expression were carried out using limma (v3.42.2), which used the Empirical Bayes method for model fitting[81]. The significantly differentially expressed genes (DEGs) were selected using a q-value (false discovery rate corrected p-value) of <0.05 and $\log_2$ fold change of >2 for upregulated and <(−2) for downregulated genes, respectively.

**Functional annotation using gene set enrichment analysis**. Identification of enriched gene sets and pathways were conducted using the Gene Set Enrichment Analysis (GSEA, v3.0) tool[82]. The gene sets used for this analysis were part of MSigDB database, which included Hallmark (50 gene sets), Canonical pathways (2199 gene sets), GO biological process (7350 gene sets), GO molecular function (1645 gene sets), oncogenic signatures (189 gene sets), immunologic signatures (browse 4872 gene sets), and chemical and genetic perturbations (3302 gene sets)[83]. Functional annotation of differentially expressed genes from tumor and normal bone comparison was done using the GSEAPreranked tool within GSEA program (FDR <0.05). In addition, we ran GSEA analysis on samples (n = 27) in the 25th and 75th percentile of DFI. Transcript expression of 15,563 genes were used as input for this analysis and enriched genes sets were identified in short (25th percentile) and long (75th percentile) DFI patients (FDR <0.05). The visualization of core enriched gene expression from selected significant gene sets were plotted as heatmaps using pheatmap (v1.0.12) R package[84].

**Immunohistochemistry**. Archived, formalin-fixed, paraffin embedded (FFPE) tissue samples were obtained from the Colorado State University Flint Animal Cancer Center Tissue Archive. Available paraffin blocks were routinely processed for hematoxylin and eosin (H&E) staining, as well as immunohistochemistry. The

H&E-stained slides were evaluated by a board-certified pathologist (DPR) to confirm diagnosis and the presence of adequate viable tumor sample for IHC analysis. Immunohistochemistry was performed via routine, automated methods on the Leica Bond Max autostainer (Leica Biosystems Inc.), with the following panel of previously published canine cross-reactive antibodies: mouse monoclonal anti-human CD3 (pan T lymphocyte marker; Leica, clone LN10, ready-to-use format), and monoclonal mouse anti-human Myeloid/Histiocyte antigen (MAC387; monocytes/macrophages; Dako, clone MAC387, 1:300 dilution/ 0.76 mg/mL). Primary antibodies were diluted in Bond ready-to-use primary antibody diluent (Leica Biosystems Inc.) and incubation was carried out at room temperature (RT) for 30 min. Antigen retrieval was performed using Leica Epitope Retrieval 2 (Tris-EDTA buffer, pH 9) for 20 min at 95°C. Detection was performed with PowerVision IHC detection systems (Leica Biosystems, Inc.), using a polymeric alkaline phosphatase anti-mouse IgG (MAC387), incubated for 25 min at RT, and Fast Red chromogen.

Whole slide brightfield images of IHC stained slides were digitally captured using an Olympus IX83 microscope at 10x magnification and fixed exposure times for all samples. Quantitative image analysis was performed using open-source ImageJ software (National Institutes of Health). Parent images were converted to gray scale.tiff images for analysis. Tumor tissue regions-of-interest (ROIs) were segmented from adjacent normal tissue, tissue section artifacts, regions of hemorrhage and necrosis, and clearly definable areas of tumor bone matrix by manual outlining in ImageJ in blinded fashion by a board-certified veterinary pathologist. Following determination of the ROI for analysis, positively labeled infiltrating immune cells were counted using the color deconvolution algorithm. Briefly, a positive pixel threshold for all immune cell markers was determined using lymph node positive control and corresponding isotype-stained control slide images and visually confirmed by a veterinary pathologist using appropriate isotype-stained control slides. Images were subjected the ImageJ to color deconvolution plug-in, followed by global, automated application of this intensity threshold to all images. Following automated image analysis, positive pixel masks of each image were blindly evaluated by a pathologist to ensure thresholding accuracy. Data was analyzed and the number of infiltrating immune cells was expressed as immune cell positive area as a percentage of total tumor tissue area. The javascript macro used for this analysis in ImageJ is included as Supplementary Note 1. Supplementary Fig. 16 provides detailed images showing the regions of exclusion and the subsequent image mask of the tumor regions that were analyzed.

**Gene expression profiling of immune cells**. In the last few years, multiple groups have characterized immune cell types using gene expression profile. Using the gene sets for immune cell types provided by Rooney et al.[46], we have calculated a score for immune cell types in each sample[46]. A score was calculated as geometric mean of gene expression for signature genes of each cell type (Supplementary Table 2). These scores were used to assess associations with DFI, mutations per megabase (tumor burden), and immune staining quantification data for T cells and myeloid cells using Pearson correlation. The significance of multiple correlations was corrected for false discovery rates using Benjamini–Hochberg method (R package fdrtool, v1.2.15). The immune cell profiling for tumor microenvironment was also carried out by using CIBERSORT tool[47]. Using transcript expression data as input, deconvolution of 22 functionally defined immune cell types was carried out[51]. A third method to calculate immune cell infiltration was done using the ESTIMATE (v1.0.13) R package for each tumor sample[48].

**Viability assays of TP53 mutant cell lines using doxorubicin**. Eight canine OSA cell lines with WT or mutant TP53[31] were assessed for sensitivity to doxorubicin. For each of 3–4 experiments, cells were plated in 96-well plates at 1000 to 6000 cells/well in complete media, treated in triplicate with serial dilutions of doxorubicin (1–500 ng/ml) or PBS vehicle and monitored using a resazurin-based fluorescence assay over 72 h. Fluorescence readings at 72 h were normalized to 0 h and expressed as the average percent of control for triplicate samples. Values (mean ± SD of 3–4 experiments) were plotted in GraphPad Prism (v9) using a sigmoidal curve fit of doxorubicin log concentration versus percent of control cell readings. The source and metadata of these eight cell lines are reported in Supplementary Table 4. These cell lines were validated using three different methods: mycoplasma testing, species PCR, and STR analysis[31,32]. A PCR-based method with an internal standard to detect mycoplasma contamination was used in each cell line[85]. Cell lines were obtained from Douglas Thamm. All cell lines used in this study tested mycoplasma free. Genomic DNA from each cell line was amplified in a multiplex PCR reaction including primers to amplify unique fragments for each of 6 commonly used species including human, cat, chinese hamster, rat, dog, and mouse[86]. All cell lines were confirmed as canine. The STR analysis was carried out using the StockMarks™ for Dogs Genotyping Kit, canine (Catalog number: 4307481) from Thermo Fisher Scientific. Cell line genomic DNA was amplified by using multiple fluorescent dye-labeled PCR primers against 10 unique microsatellite markers (STRs) simultaneously in a multiplex PCR reaction. The resulting PCR products were analyzed using fragment analysis with size standards. Identified peaks were hand-binned and compared to 65 previously analyzed canine cell lines[87].

**Statistics and reproducibility**. The association of *TP53* mutation status, monocyte counts, and IHC staining scores with patient DFI was assessed using Kaplan–Meier survival analyses (R packages used: survival, v.3.2-3 and survminer, v.0.4.8). The univariate Cox proportional hazard regression (R package: survival, v.3.2-3) was used to identify significant association of clinical parameters and *TP53* mutation status to patient outcome. A *p* value cut-off of 0.2 was used to select factors for the stepwise algorithm. The stepwise forward multivariate COXPH regression was carried out using R package My.stepwise (v0.1.0). The significance level for model entry and exit was 0.1. The final model was comprised of factors what were significantly associated with outcome at a *p*-value less than 0.05. For multiple testing FDR <0.05 was considered significant.

The whole exome data was processed on the RMACC Summit supercomputer[88]. The tools, databases, and their versions, in addition to the bioinformatic pipeline used in this study can be accessed here: https://github.com/sdas2019/Canine-Osteosarcoma-Whole-Exome-Sequencing-Pipeline.

**Ethics approval and consent to participate**. The animal study was reviewed and approved by Colorado State University Animal Care and Use Committee. Written informed consent was obtained from the owners for the participation of their animals in this study.

**Reporting summary**. Further information on research design is available in the Nature Research Reporting Summary linked to this article.

## Data availability

The Illumina raw fastq files were submitted to NCBI Short Read Archive (SRA) database. The canine osteosarcoma and normal samples have been submitted to Bioproject PRJNA613479 and PRJNA503860, respectively. The microarray data can be downloaded from Gene Expression Omnibus (GEO) database using accession number GSE76127 and GSE180303. All processed data are available within this article or as supplementary data and information. Source data for plots are provided in Supplementary Data 11.

## Code availability

The javascript macro used to measure the immunohistochemical sections to determine the percentage of tumor stained with antibodies in ImageJ is included in Supplementary Note 1.

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

## Acknowledgements

We thank Tyler Eike for IT support and software installation. We would also like to thank Keegan Collins for prepping tumor samples, Liza Pfaff for prepping normal bone samples, and Irene Mok for her help with sample curation. The authors gratefully acknowledge the efforts of Mary Lafferty for collection of clinical outcome data. Funding was provided by Anschutz Foundation (DLD, DLG and DHT), P30 CA046934 (University of Colorado Cancer Center Support Grant, Genomics and Microarray Shared Resource and DL Gustafson). Funding was also provided by COXEN Retrospective Grant: Morris Animal Foundation D13CA-044 (PI: DL Gustafson) and COXEN Prospective Grant: Morris Animal Foundation D16CA-003 (PI: DL Gustafson). Additional funding includes the National Institutes of Health, Office of the Director, award number K01ODO22982 (PI: DP Regan) and National Center for Advancing Translational Sciences, award number L30 TR002126 (PI: DP Regan).

## Author contributions

S.D. and D.L.D. contributed towards design of the work, analyses, and interpretation of data, and drafting and revisions of the manuscript. R.I. and J.S.F. was responsible for sample preparation used for whole exome sequencing and microarray, respectively. D.P.R. contributed towards immunohistochemical sample processing, interpretation, and analyses of immunogenomic analyses. S.E.L. contributed towards curations of tumor samples and is the Director of FACC tissue archive. D.H.T. contributed towards analysis and interpretation of survival analysis and revision of the draft. D.L.G. contributed towards providing resources for gene expression analysis and interpretation.

## Competing interests
The authors declare no competing interests.
