## [Peer Review File · Communications Biology]

Reviewers' comments:

Reviewer #1 (Remarks to the Author):

This is a well written and comprehensive study relating to 26 cases of osteosarcoma in dogs. The importance of a comparative model has been made clear and this paper brings a significant amount of data into the public domain. It is a comprehensive and descriptive data set that also links back to survival times (although in a limited number of cases). Fundamentally, there is no issue with methodology or interpretation. What this lacks is functional analysis following the descriptive analysis, although this is an editorial decision of whether this is an issue. Otherwise, I would recommend publication.

Reviewer #2 (Remarks to the Author):

Das et al present a study of 26 canine osteosarcoma cases and matched normal tissue, including whole exome sequencing, RNA expression by array, and immunohistochemistry. Extensive bioinformatic analyses were used to identify common variants and pathways with altered expression. Recurrent mutations were found in TP53 and SETD2, which confirms previous studies, and the high degree of copy number variants versus SNVs and INDELS found is also in line with what is already known about canine osteosarcoma. The bioinformatic portions of this study are well-conducted and analyses are appropriate. While this data confirms and builds on previous studies, it does not provide much novel information.

The major new findings in this study are based on the associations between clinical outcome and several measurements: presence of TP53 mutations, expression changes in various gene sets (especially ECM, drug metabolism and immune response), and immune cell infiltration into tumours. While this information is novel, the methodology used to define clinical outcome and immune cell infiltration is not clear and appears to be biased or incorrect.

Major issues:

1. A fundamental component of this study is the use of DFI to examine correlations between mutation and expression data and clinical outcome. Since only the top and bottom quartiles are compared this means that all the clinical correlations in this study hang on just 12 dogs. As such, much more detail on how the authors calculated DFI is required.

A) First, inclusion criteria must be stated in much more detail. As far as I can see, all the dogs received amputation surgery and chemotherapy and they all received at least one dose of chemotherapy. While it is likely that all dogs at minimum had thoracic radiographs for staging before treatment to make sure there were no lung metastases present, this needs to be explicitly stated. i.e. the presence of lung metastasis pre-treatment must be stated as an exclusion criterion. The timing of chemotherapy treatment, both post-surgery and between each chemo treatment is likely to be slightly variable, but this information would be helpful to the reader and could be included as supplemental information. A patient that had a low WBC count that delayed chemotherapy is not likely to have a robust immune response to micro-metastatic lesions, yet that is the basis for the mechanism proposed in this paper for longer DFI.

B) Second, and most importantly, the determination of the date when disease was deemed to have progressed must be defined very precisely. As the authors state themselves on page 23, line 466, when discussing another study, "In addition, survival times can be confounded by the use of euthanasia in companion animals, making DFI a potentially more biologically relevant measure of outcome." But this does not necessarily make DFI a better measure of clinical outcome. As the authors point out, canine osteosarcoma affects older large and giant breed dogs. These dogs are likely to have comorbidities – osteoarthritis and cardiovascular disease being common. Determining if an osteosarcoma patient has progression of disease means detection of metastasis as the primary tumour almost never recurs at the amputation site. Canine osteosarcoma metastasizes most frequently to bones and lungs. A limping dog (arthritis? injury? or metastasis?) or a dog with respiratory signs (heart disease? lung disease? metastasis?) could be counted as having metastasis and even euthanized, especially if they have other comorbidities such as endocrine disorders or depending on such non-disease-related factors as the owner's financial

situation. These older patients are at risk of developing other malignancies that originate in or metastasize to lung as well, and thus may be mistaken for osteosarcoma progression.

The authors must state very precisely how they determined that a patient had progressed and developed osteosarcoma metastasis. The only sure way is to have histopathologic confirmation of metastatic osteosarcoma, but this is not commonly pursued. Radiographic evidence of lung or bone metastasis is a reasonable proxy, though there will still be some false positives. As outlined above, lameness/bone pain and respiratory signs are commonly attributed to metastasis but are much less specific. In a study such as this where comparisons are made between 2 groups of only 6 patients (the long and short DFI), removing a patient or 2 can drastically change the statistical analysis and significance of the findings. Did all the dogs in the study have serial radiographs at specific intervals to establish DFI? Did all the dogs that had clinically suspected progression have it verified by radiographs? Occasionally, dogs with radiographic evidence of lung metastasis survive many months without any signs of the disease, yet their DFI ends up defined as short. How were these cases handled? The authors should consider including disease-specific survival and overall survival in the analyses. This may not yield significant associations with p53 status or immune infiltration, but as long as the criteria are precisely defined, it adds to our understanding of the disease.

The issue of determining time to metastasis and tumour-specific mortality (overall versus death due to progressive disease) issue has been highlighted recently in the veterinary literature in an editorial in *Veterinary Pathology* (Meuten, Munday, and Hauck, *Vet Pathol.* 2018 Mar;55(2):195-199. doi: 10.1177/0300985817753869.) It is critical that information of how progressive disease was determined is included for each case.

Related to this, Page 15, line 329: The DFI of the patient alive on July 23, 2020 should be updated.

2. IHC and quantification of immune cell infiltrates:

Osteosarcoma tumours have variable mineralization. Some require demineralization in (usually) acid solutions to allow histological sectioning, either through soaking the whole sample or surface demineralizing when sectioning. However, some tumours (or regions of tumours) do not require this for sectioning, and since this is a faster process (and it is easier to obtain this tissue from a biopsy because it is softer), there is a potential bias towards sectioning and analyzing less mineralized pieces of tumour. This has several problematic effects.

A) Acid treatment can affect antigen conformation and IHC. Were some samples demineralized and some not? Were some surface demineralized and some soaked? (even soaking time is likely going to vary from tumour to tumour, often by several days.) Are the IHC protocols the authors used affected by demineralization? (In this reviewer's experience, the effect of decal solutions on IHC is highly variable and unpredictable from one antigen-antibody protocol to another.) If demineralization affects IHC for the immune cells the authors quantified, then there is a potential bias towards finding more (or less) in tumours that were more or less mineralized.

B) Irrespective of the above potential IHC issues, the authors quantified immune infiltrates based on percentage of total tumour tissue area. This is not explained in enough detail for anyone to replicate and may not be valid. As mentioned above, osteosarcomas have highly variable amounts of stroma, as well as necrosis, hemorrhage, and other non-tumour cell components. Raw surface area on a slide does not take this into account even if surrounding non-tumour tissue is excluded. For any given % of tumour area, osteosarcomas with more osteoid/stroma are going to have more immune cells per tumour cell than those with less osteoid. Since the tumour cells are the target of the immune infiltrate, not the stroma, this will produce biased results that are heavily in favour of tumours with a large non-tumour cell component having LESS immune cells as a % of tumour area but MORE immune cells per tumour cell.

To correct these biases, the authors must first provide more details on how they quantified tumour area and then be sure to quantify immune infiltrates either based on area of immune cells as a % of tumour cell area, or on a % immune cells per tumour cell (i.e. counting cells rather than using area).

3. Comparison to other studies:

A) There are a few other canine osteosarcoma studies that should be included in the discussion. Shao et al, *Cross-species genomics identifies DLG2 as a tumor suppressor in osteosarcoma*

Oncogene. 2019 Jan;38(2):291-298. doi: 10.1038/s41388-018-0444-4. This study found DLG2 deleted in over half of canine osteosarcomas, then compared this to human data (42% had lost it) and also confirmed a mechanism using mouse models. DLG2 is listed in Table S5 as significantly deleted but is only mentioned in the manuscript in the context of human osteosarcoma, not canine, and the % of cases in the current study that had this deletion is not stated. More information and discussion of DLG2 loss would be useful for the audience.

Ayers et al. Novel application of single-cell next-generation sequencing for determination of intratumoral heterogeneity of canine osteosarcoma cell lines. J Vet Diagn Invest. 2021. doi: 10.1177/1040638720985242. This study came out after submission of the current study, but it would be worth some discussion in a revised manuscript as the transcriptomic analysis is similar and they found distinct clusters across canine osteosarcoma cell lines. This would be an important comparison for the research community with respect to how well these cell lines model actual disease.

Likewise, Simpson et al. Cancers Molecular Characterisation of Canine Osteosarcoma in High Risk Breeds Cancers (Basel). 2020 Aug 25;12(9):2405. doi: 10.3390/cancers12092405. has similar expression analyses and deserves comparison, especially to the Rottweilers and Irish Wolfhounds in the current study.

Minor points:

1. A very recent paper (published after this study was submitted) is highly supportive of the idea that mutant p53 alters immune signaling and promotes tumorigenesis. The authors should consider adding this reference. (Ghosh M et al, Mutant p53 suppresses innate immune signaling to promote tumorigenesis. Cancer Cell 2021 Jan 23;S1535-6108(21)00041-6. doi: 10.1016/j.ccell.2021.01.003.)

2. Page 5 line 107: The first quartile of DFI defined here as <123 days and the authors state that tumours from this quartile were enriched for genes in immune-related pathways. But in the paragraph that begins on page 14, line 297 they state the opposite.

3. Page 36, IHC: There are a few details lacking from these methods that would make it difficult for someone to repeat the experiments: concentration of antibodies, diluent used, incubation and antigen retrieval temperatures, incubation times. These could be placed in a supplemental methods section, but they are important for someone to be able to repeat the findings or compare to their own findings.

4. Table 2A and 2B. Some p values are stated as "0" or "0.000". The stats program may spit out this value, but it should be reported as the actual p value, using an exponent if necessary (as in Table S6C), or being less than a predetermined cutoff, e.g. 0.0001.

5. Table S1A: The column "In-house sequencing project name" is meaningless to the reader and is confusing. There are patient names ("Bella", "Rocky", etc.) that, when paired with all the other data, clearly identify the individual and potentially the owner/client. These should be removed. Likewise, Table S1B includes date of birth, but this not used in the analysis, and is irrelevant and should be removed.

Reviewer #3 (Remarks to the Author):

Osteosarcoma is a rare tumor in the human species being is the sixth leading cause of malignancy lesion in children under 15, however in dogs it occurs at any age as well as in any breed, but it habitually develops in older, large and giant breed. Thus, if we look at the incidence rate in canine specie it can be 27 times higher. The prognosis for both species is relatively poor, in dogs, the 1-year survival rates can be considered low, so requiring new therapeutic methods that can improve the survival of these patients. The information from genetic studies can dramatically help in the improvement of therapeutic methods to benefit both species, with the advantage that the incidence rate of these neoplasms being higher in canines can contribute as an excellent disease model with benefit for both species.

I recommend that the authors check the legend of figure 4 that I imagine was replaced with the figure 5, not corresponding to the content of the immunohistochemistry images documented there.

Reviewer #1 (Remarks to the Author):

This is a well written and comprehensive study relating to 26 cases of osteosarcoma in dogs. The importance of a comparative model has been made clear and this paper brings a significant amount of data into the public domain. It is a comprehensive and descriptive data set that also links back to survival times (although in a limited number of cases). Fundamentally, there is no issue with methodology or interpretation. What this lacks is functional analysis following the descriptive analysis, although this is an editorial decision of whether this is an issue. Otherwise, I would recommend publication.

Our response:

We appreciate the reviewer's comment that the study lacks functional analysis to support the associations identified in this study. Although the samples included in this study were archived from prior patients, we have provided additional discussion and analysis of available canine osteosarcoma cell lines to help provide functional analysis for these findings. To support the hypothesis that missense mutations of *TP53* may increase sensitivity of canine osteosarcomas to DNA damaging chemotherapeutics as previously described, we have tested the relative sensitivity of 8 canine osteosarcoma cell lines to doxorubicin (Supplementary Figure 14). We found that the HMPOS cell line with a *TP53* R175H homolog variant (R226H) as a potential driver, also exhibited the greatest sensitivity to doxorubicin among the 8 canine osteosarcoma cell lines tested. This analysis included two other cell lines with *TP53* missense mutations (R301W/C329F and I284T) and 5 *TP53* wildtype cell lines. We also provided additional references to indicate that exon 6 truncated p53 may have unique gain of function activities that increase metastatic potential. In support of this, we found that Kaplan-Meier survival analysis indicated that tumors with exon 6 truncations (homologous to human variants hI195*, hR196*, and hR213*) as their major *TP53* mutations, also had significantly shorter disease-free intervals than patients with either wildtype or missense mutated *TP53* bearing tumors (Supplementary Figure 15). None of the available canine cell lines had truncating mutations in *TP53*.

We have added the following statements to the manuscript (see lines 490-516).

*“Using overall survival as the time event in Kaplan-Meier analysis, dogs with *TP53* missense mutations continued to have statistically better outcomes than the wildtype/ truncated group (Supplementary Figure 13). A study in human small-cell lung cancer associated the *TP53* mutations identified in 54% of patients with longer relapse free intervals compared to patients with wild type *TP53*⁵⁵. Similarly, *TP53* mutant human cancers including breast, are significantly more likely to achieve pathological complete responses to chemotherapy⁵⁶⁻⁶⁰. Examination of this phenomenon has shown that tumors from mice with the murine *Tp53* R172H variant exhibit greater sensitivity to doxorubicin, and fail to exit the cell cycle following treatment, resulting in aberrant mitosis, and cell death⁶¹. Similarly, we found that among a panel of canine OSA cell lines, the HMPOS cell line which bears the homologous R to H variant³², also had the greatest sensitivity to doxorubicin (Supplementary Figure 14). Human *TP53* mutants, including R175H, R248W, and R273H, inactivate the ATM-dependent DNA damage response leading to chromosomal translocations and a defective G2/M checkpoint, and improved treatment responses⁶². A similar phenomenon may occur in *TP53* mutant canine OSA resulting in a longer*

DFI following chemotherapy in dogs bearing missense mutations in TP53. Conversely, forced expression of WT-p53 in p53 null A549 human lung carcinoma cells promoted cell survival in response to double strand break-inducing agents like doxorubicin by promoting cell cycle arrest and DNA damage repair⁶³. Further, for this analysis, we grouped tumors with WT p53 with those bearing TP53 truncations to increase the numbers of tumors that lacked missense mutations. However, these TP53 truncations (I247, R248*, R265*) are homologous to human exon 6 truncations I195*, hR196*, and hR213* and have the shortest DFIs among the WES group (Supplementary Figure 15). In human cancers, exon 6 truncations have no nuclear transcription regulatory activity, instead they are localized to the mitochondria and interact with cyclophilin to regulate the mitochondrial permeability transition pore. Exon 6 truncated TP53 increases reactive oxygen generation, epithelial to mesenchymal transition, and drives lung metastasis in melanoma cells^{64,65}. Interestingly, there are 8 variants localized to AA245-265, including 5 additional missense mutations (Supplementary Figure 6).”*

Reviewer #2 (Remarks to the Author):

Das et al present a study of 26 canine osteosarcoma cases and matched normal tissue, including whole exome sequencing, RNA expression by array, and immunohistochemistry. Extensive bioinformatic analyses were used to identify common variants and pathways with altered expression. Recurrent mutations were found in TP53 and SETD2, which confirms previous studies, and the high degree of copy number variants versus SNVs and INDELs found is also in line with what is already known about canine osteosarcoma. The bioinformatic portions of this study are well-conducted and analyses are appropriate. While this data confirms and builds on previous studies, it does not provide much novel information.

The major new findings in this study are based on the associations between clinical outcome and several measurements: presence of TP53 mutations, expression changes in various gene sets (especially ECM, drug metabolism and immune response), and immune cell infiltration into tumours. While this information is novel, the methodology used to define clinical outcome and immune cell infiltration is not clear and appears to be biased or incorrect.

Major issues:

1. A fundamental component of this study is the use of DFI to examine correlations between mutation and expression data and clinical outcome. Since only the top and bottom quartiles are compared this means that all the clinical correlations in this study hang on just 12 dogs. As such, much more detail on how the authors calculated DFI is required.

Our response:

To address the low number of samples for this section of the paper, we have added microarray data from a total of 108 OSA samples. After eliminating the censored samples, the sample size in bottom and upper quartiles was increased to 27 samples (See Supplementary Table 1C). For detailed response on DFI calculations please refer to the next response.

A) First, inclusion criteria must be stated in much more detail. As far as I can see, all the dogs

received amputation surgery and chemotherapy and they all received at least one dose of chemotherapy. While it is likely that all dogs at minimum had thoracic radiographs for staging before treatment to make sure there were no lung metastases present, this needs to be explicitly stated. i.e. the presence of lung metastasis pre-treatment must be stated as an exclusion criterion. The timing of chemotherapy treatment, both post-surgery and between each chemo treatment is likely to be slightly variable, but this information would be helpful to the reader and could be included as supplemental information. A patient that had a low WBC count that delayed chemotherapy is not likely to have a robust immune response to micro-metastatic lesions, yet that is the basis for the mechanism proposed in this paper for longer DFI.

B) Second, and most importantly, the determination of the date when disease was deemed to have progressed must be defined very precisely. As the authors state themselves on page 23, line 466, when discussing another study, “In addition, survival times can be confounded by the use of euthanasia in companion animals, making DFI a potentially more biologically relevant measure of outcome.” But this does not necessarily make DFI a better measure of clinical outcome. As the authors point out, canine osteosarcoma affects older large and giant breed dogs. These dogs are likely to have comorbidities – osteoarthritis and cardiovascular disease being common. Determining if an osteosarcoma patient has progression of disease means detection of metastasis as the primary tumour almost never recurs at the amputation site. Canine osteosarcoma metastasizes most frequently to bones and lungs. A limping dog (arthritis? injury? or metastasis?) or a dog with respiratory signs (heart disease? lung disease? metastasis?) could be counted as having metastasis and even euthanized, especially if they have other comorbidities such as endocrine disorders or depending on such non-disease-related factors as the owner’s financial situation. These older patients are at risk of developing other malignancies that originate in or metastasize to lung as well, and thus may be mistaken for osteosarcoma progression.

The authors must state very precisely how they determined that a patient had progressed and developed osteosarcoma metastasis. The only sure way is to have histopathologic confirmation of metastatic osteosarcoma, but this is not commonly pursued. Radiographic evidence of lung or bone metastasis is a reasonable proxy, though there will still be some false positives. As outlined above, lameness/bone pain and respiratory signs are commonly attributed to metastasis but are much less specific. In a study such as this where comparisons are made between 2 groups of only 6 patients (the long and short DFI), removing a patient or 2 can drastically change the statistical analysis and significance of the findings. Did all the dogs in the study have serial radiographs at specific intervals to establish DFI? Did all the dogs that had clinically suspected progression have it verified by radiographs? Occasionally, dogs with radiographic evidence of lung metastasis survive many months without any signs of the disease, yet their DFI ends up defined as short. How were these cases handled? The authors should consider including disease-specific survival and overall survival in the analyses. This may not yield significant associations with p53 status or immune infiltration, but as long as the criteria are precisely defined, it adds to our understanding of the disease.

The issue of determining time to metastasis and tumour-specific mortality (overall versus death due to progressive disease) issue has been highlighted recently in the veterinary literature in an editorial in *Veterinary Pathology* (Meuten, Munday, and Hauck, *Vet Pathol.* 2018 Mar;55(2):195-199. doi: 10.1177/0300985817753869.) It is critical that information of how

progressive disease was determined is included for each case.

Our response:

We appreciate the reviewer's thoughtful explanation of the importance of carefully delineated criteria for study inclusion and evaluation of disease progression. To address this issue, we have added the following statements regarding the inclusion/exclusion criteria for the patients in the method section titled: "**Sample selection for retrospective study**" (see lines 575-583).

"Primary tumors were selected from dogs meeting the following criteria: diagnosed with osteosarcoma and receiving treatment consisting of limb amputation and chemotherapeutic treatment protocols including at least one dose of platinum-based or doxorubicin-based chemotherapy or both. For inclusion in this retrospective cohort study, dogs must have had pre-operative thoracic radiographs or computed tomography and histopathologic confirmation of the diagnosis. Exclusion criteria were presence or suspicion of metastases at any site before amputation or prior treatment of appendicular OSA with radiation (palliative or curative-intent) protocols, chemotherapy, or surgery."

In addition, we provide the following details for the reviewer's information. For the 26 patients included in both the WES and microarray analysis, initial staging was done with three view chest radiographs in all cases. In addition, 17 cases had staging with whole body bone scans using Technicium-99. In addition to chest radiographs, one patient had a bone scan and thoracic CT. No patient was suspected to have metastatic disease based on staging tests at the time of amputation.

In regard to the identification of disease progression, we have added the following statement (see lines 584-589).

"In most cases patients were monitored for disease progression and metastasis with physical exam and chest radiographs at 3-month intervals or sooner if clinical symptoms prompted additional screening. Pulmonary lesions were considered metastasis if they were multiple, progressive on serial radiographs, or confirmed on necropsy. Non pulmonary lesions were considered metastatic if they were suspicious for neoplasia based on radiographic or CT appearance or confirmed with histopathology either at the time of surgical removal or based on necropsy."

For the reviewer's information we provide the following details. For the 26 patients included in the WES analysis, 14 patients were considered progressive based on only radiographic evidence of pulmonary or secondary bone lesions, 2 patients had disease progression confirmed via histopathology of suspicious lesions and 4 had necropsy confirmation of progressive lesions at the time of death. Three patients died or were euthanized due to non-osteosarcoma causes (GDV, Sepsis, orthopedic failure of limb) without evidence of OSA progression at the time of death based on routine radiographic monitoring. One patient was lost to follow up with no evidence of metastasis at the time of last contact 90 days post amputation, this patient was censored from the DFI groups. One patient (T-29C) was still alive as of 6/25/21 with no evidence of disease

progression based on routine thoracic radiographic monitoring. Cause of death in one patient (T-1246) was unknown but was presumed DOD at 756 days post diagnosis and treatment.

In regard to the chemotherapeutic treatment of the WES patients and the expanded microarray analysis group, we have included the following statement in the materials and methods (see lines 591-597).

“Chemotherapeutic treatment of the 109 patients following amputation is detailed in Table S1A. Carboplatin as a single agent was administered using a protocol of 300 mg/m² given every 21 days for 4 or 6 treatments. Doxorubicin as a single agent was administered at 30mg/m² every 21 days for 5 treatments. Alternating carboplatin and doxorubicin using the doses listed above, occurred every 21 days for a total of 6 treatments, 3 of each drug. Three dogs included in the microarray analysis received subcutaneous injections of a slow-release cisplatin polymer mixture (Atrigel) at a dose of approximately 60 mg/m².”

For the reviewer’s information, we provide the following information regarding dose delays or reductions. Of the 26 WES patients, 14 (54%) completed their prescribed chemotherapy protocol. Protocols were not complete in 8 patients due to progression of disease prior to protocol completion and 4, due to owners electing to discontinue treatment. Three of 26 patients (11%) had a dose reduction due to hematologic or gastrointestinal toxicity, one of these patients was censored at 77 days from non-disease related death. Eight of 26 patients (31%) had a dose delay, 4 due to toxicity or secondary infection and 4 due to owner scheduling (2 DFI<90 days, 2 DFI> 458 days).

As the reviewer pointed out, with this small group of osteosarcomas, there is a much larger chance that delayed diagnosis of metastasis in even one patient can have a large impact on the differential gene expression analysis. Although we were unable to enlarge our WES group, we have extended the GSEA by analyzing 108 dogs (including the 26 WES dogs, Supplementary Table 1C) meeting the previously described inclusion and exclusion criteria whose tumors were analyzed using Affymetrix Canine 2.0 microarrays from 3 studies. Exploring GSEA analysis in the lowest and highest DFI quartiles allows us to compare expression profiles for 27 dogs with DFI <90 days compared to 27 dogs with DFI >458 days. This expanded data set allowed us to identify 37 and 17 pathways/terms were enriched in long and short DFI patients, respectively (**Table 1 and Supplementary Table 9**). The gene sets upregulated in short DFI patients included cell cycle pathways, DNA repair pathways, and MYC and E2F targets. The Hallmark gene sets upregulated in long DFI patient tumors included several immune response datasets: Interferon alpha and gamma response, inflammatory response, and allograft rejection along with ECM gene sets (**Supplementary Table 9, Supplementary Figure 10**). Thus, we found that immune response data sets continued to play a significant role in those patients with a long disease-free interval, while the specific data sets enriched in short DFI patients has expanded to include more generalized mechanisms to drive cellular proliferation and repair of DNA damage rather than the more specific pathways previously identified with the smaller dataset. We appreciate the reviewer’s comment and feel that this expanded analysis has much greater relevance (see lines 322-348 in manuscript).

Related to this, Page 15, line 329: The DFI of the patient alive on July 23, 2020 should be updated.

As mentioned in the Supplementary Table 1C and in text, we have updated the DFI for the living patient (1533 days; See line 376). This resulted in updating of the Kaplan Meier plots (see Fig. 6, and Supplementary figure 11).

2. IHC and quantification of immune cell infiltrates:

Osteosarcoma tumours have variable mineralization. Some require demineralization in (usually) acid solutions to allow histological sectioning, either through soaking the whole sample or surface demineralizing when sectioning. However, some tumours (or regions of tumours) do not require this for sectioning, and since this is a faster process (and it is easier to obtain this tissue from a biopsy because it is softer), there is a potential bias towards sectioning and analyzing less mineralized pieces of tumour. This has several problematic effects.

A) Acid treatment can affect antigen conformation and IHC. Were some samples demineralized and some not? Were some surface demineralized and some soaked? (even soaking time is likely going to vary from tumour to tumour, often by several days.) Are the IHC protocols the authors used affected by demineralization? (In this reviewer's experience, the effect of decal solutions on IHC is highly variable and unpredictable from one antigen-antibody protocol to another.) If demineralization affects IHC for the immune cells the authors quantified, then there is a potential bias towards finding more (or less) in tumours that were more or less mineralized.

Our response:

The tissue archiving staff at the CSU-Flint Animal Cancer Center uses a SOP for formalin fixation of patient-derived tumor samples. For osteosarcoma biopsies, these samples do not undergo any demineralization treatment. Tumor biopsies are able to be obtained/cut with a scalpel or trephine. As the reviewer notes, this could introduce potential bias to the sample set as only softer tumor tissues with a lesser degree of mineralization are analyzed; however, as the reviewer also notes, there is really no way around this bias, as the effect of decal solutions on antigen preservation and immunoreactivity is highly variable and unpredictable, precluding reliable IHC-based immune profiling of demineralized canine osteosarcoma tumor biopsies. Additional information on antibody dilution and incubation period have been reported in lines 685-695.

“Immunohistochemistry was performed via routine, automated methods on the Leica Bond Max autostainer (Leica Biosystems Inc.), with the following panel of previously published canine cross-reactive antibodies: mouse monoclonal anti-human CD3 (pan T lymphocyte marker; Leica, clone LN10, ready-to-use format), and monoclonal mouse anti-human Myeloid/Histiocyte antigen (MAC387; monocytes/macrophages; Dako, clone MAC387, 1:300 dilution/0.76 mg/mL). Primary antibodies were diluted in Bond ready-to-use primary antibody diluent (Leica Biosystems Inc.) and incubation was carried out at room temperature (RT) for 30 min. Antigen retrieval was performed using Leica Epitope Retrieval 2 (Tris-EDTA buffer, pH 9) for 20 min at 95°C. Detection was performed with PowerVision IHC detection systems (Leica Biosystems,

Inc.), using a polymeric alkaline phosphatase anti-mouse IgG (MAC387), incubated for 25 min at RT, and Fast Red chromogen.”

B) Irrespective of the above potential IHC issues, the authors quantified immune infiltrates based on percentage of total tumour tissue area. This is not explained in enough detail for anyone to replicate and may not be valid. As mentioned above, osteosarcomas have highly variable amounts of stroma, as well as necrosis, hemorrhage, and other non-tumour cell components. Raw surface area on a slide does not take this into account even if surrounding non-tumour tissue is excluded. For any given % of tumour area, osteosarcomas with more osteoid/stroma are going to have more immune cells per tumour cell than those with less osteoid. Since the tumour cells are the target of the immune infiltrate, not the stroma, this will produce biased results that are heavily in favour of tumours with a large non-tumour cell component having LESS immune cells as a % of tumour area but MORE immune cells per tumour cell.

To correct these biases, the authors must first provide more details on how they quantified tumour area and then be sure to quantify immune infiltrates either based on area of immune cells as a % of tumour cell area, or on a % immune cells per tumour cell (i.e. counting cells rather than using area).

Our response:

Additional detail has been added to the quantitative image analysis section of the manuscript methods on lines 697-714. Additionally, we have also provided the ImageJ macro javascript with all the necessary code language (see Supplementary Note) so that any other investigator could import this code to freely available Image J software and repeat the analysis themselves, utilizing this fully automated script. For the data presented in this manuscript, the pathologist performed significant quality control evaluation and verification of tumor regions-of-interest (ROIs) pre- and post-quantitative image analysis. Specifically, the pathologist ensured exclusion of all normal tissue from this analysis including adjacent normal tissue, reactive fibroplasia, and reactive bone, all hemorrhage, and all necrosis. Additionally, all clearly definable regions of tumor bone matrix were excluded in our analysis to the best of our abilities. Admittedly, there are of course thin foci and trabeculae of osteoid throughout the tumor mass which are so intimately associated with osteosarcoma tumors that they cannot be either computer recognized and or removed through manual annotation. The authors feel attempting to remove these microscopic foci of osteoid would be the equivalent of this reviewer asking us to go in and exclude every other non-malignant cell type, and a counter argument could be made that one should exclude blood vessels/endothelial cells, intra-tumoral fibroblasts, any other immune cell type besides the one being quantified (i.e. exclude macrophages in the %CD3 area data), and other intimately associated aspects of the tumor stroma. In immunohistochemically quantifying the immune landscape of human pancreatic ductal adenocarcinoma, one would not exclude the desmoplastic stroma that accounts for 90%+ of the tumor mass.

The strongest piece of data supporting the accuracy of this quantitative image analysis is the statistically significant and strong linear correlation between CD3+ cell density as determined by IHC and CD3e, and specifically CD8a, mRNA expression (insert $r=0.94$ and q value= $3.3e-07$). Thus, the quantitative image analysis data quantifies immune cell positive area as a percentage of

tumor cell area only, which in the authors' opinion was done to the best ability of any pathologist and quantitative image analysis software, and much better and more detailed standard than what is typically published in the field (i.e. a semi-quantitative scoring field or counting immune cells in a limited number of pathologist-selected high magnification fields).

In supplementary Figure 16, we have provided detailed images as supplemental data showing the regions of exclusion and the subsequent image mask of the tumor regions that were analyzed.

Excerpt from manuscript (lines 695-719):

“Whole slide brightfield images of IHC stained slides were digitally captured using an Olympus IX83 microscope at 10x magnification and fixed exposure times for all samples. Quantitative image analysis was performed using open-source ImageJ software (National Institutes of Health). Parent images were converted to gray scale .tiff images for analysis. Tumor tissue regions-of-interest (ROIs) were segmented from adjacent normal tissue, tissue section artifacts, regions of hemorrhage and necrosis, and clearly definable areas of tumor bone matrix by manual outlining in ImageJ in blinded fashion by a board-certified veterinary pathologist. Following determination of the ROI for analysis, positively labeled infiltrating immune cells were counted using the color deconvolution algorithm. Briefly, a positive pixel threshold for all immune cell markers was determined using lymph node positive control and corresponding isotype-stained control slide images and visually confirmed by a veterinary pathologist using appropriate isotype-stained control slides. Images were subjected to the ImageJ color deconvolution plug-in, followed by global, automated application of this intensity threshold to all images. Following automated image analysis, positive pixel masks of each image were blindly evaluated by a pathologist to ensure thresholding accuracy. Data was analyzed and the number of infiltrating immune cells was expressed as immune cell positive area as a percentage of total tumor tissue area. The javascript macro used for this analysis in ImageJ is included as Supplementary Note. The Supplementary Figure 15 provides detailed images showing the regions of exclusion and the subsequent image mask of the tumor regions that were analyzed.”

3. Comparison to other studies:

A) There are a few other canine osteosarcoma studies that should be included in the discussion. Shao et al, Cross-species genomics identifies DLG2 as a tumor suppressor in osteosarcoma *Oncogene*. 2019 Jan;38(2):291-298. doi: 10.1038/s41388-018-0444-4. This study found DLG2 deleted in over half of canine osteosarcomas, then compared this to human data (42% had lost it) and also confirmed a mechanism using mouse models. DLG2 is listed in Table S5 as significantly deleted but is only mentioned in the manuscript in the context of human osteosarcoma, not canine, and the % of cases in the current study that had this deletion is not stated. More information and discussion of DLG2 loss would be useful for the audience.

Our response:

We have reviewed our data and incorporated the percentage of samples with significantly deleted or amplified CNVs in Supplementary Table 5. In addition, we have added the following discussion of DLG2 at lines 267-270.

“Previous studies in human and canine OSA have identified ATRX and DLG2 as SV or CNV^{7,12}. Similarly, we identified DLG2 copy number losses in 62% of our OSA samples, however, the CNV values did not significantly correlate with DLG2 transcript expression ($R=0.21$, $p=0.3$).”

Ayers et al. Novel application of single-cell next-generation sequencing for determination of intratumoral heterogeneity of canine osteosarcoma cell lines. J Vet Diagn Invest. 2021. doi: 10.1177/1040638720985242. This study came out after submission of the current study, but it would be worth some discussion in a revised manuscript as the transcriptomic analysis is similar and they found distinct clusters across canine osteosarcoma cell lines. This would be an important comparison for the research community with respect to how well these cell lines model actual disease.

Our response:

We appreciate the comment about comparison with sc-RNA-seq article on canine osteosarcoma cell lines. However, meaningful comparison between sc-RNA-seq data from two OSA cell lines and microarray data using bulk RNA from 108 tumor samples is beyond the scope of this paper.

However, we were able to compare individual gene expression of the cell lines to the tumors. We have added the following at lines 313-329.

“A recent article reported loss of PTEN expression in canine OSA cell lines (POS/HMPOS) and expression of RB1 in all four tested cell lines³¹. Although, TP53 mutation status was not reported in this study, WES analysis of 8 canine OSA cell lines coupled with prior microarray analysis, identified TP53 missense mutations in 3 cell lines (Supplementary Figure 14), and reduced RB1 expression in 2 cell lines^{32,33}. It also confirmed the loss of PTEN in HMPOS cell line, in addition to OSA8 and Abrams cell lines, and identified elevated MDM2 expression in D17 cell line³². These gene expression variations have also been reported in both canine and human OSA tumors.”

Likewise, Simpson et al. Cancers Molecular Characterisation of Canine Osteosarcoma in High Risk Breeds

Cancers (Basel). 2020 Aug 25;12(9):2405. doi: 10.3390/cancers12092405. has similar expression analyses and deserves comparison, especially to the Rottweilers and Irish Wolfhounds in the current study.

Our response:

Thank you for directing us to this article. We have compared our relevant data with those reported in the article and have added the following to our discussion at lines 307-311.

“Further, RNAseq analysis comparing the gene expression profiles between 4 normal and 3 OSA samples similarly identified dysregulated muscle and muscle contraction related pathways as well as iron homeostasis and extracellular matrix genes³⁰. Among the genes that they confirmed using RT-qPCR and IHC analysis, we also saw elevated expression of MMP3, SLC2A1, DKK3, POSTN and ASPN in OSA tumors compared to normal bone samples.”

Minor points:

1. A very recent paper (published after this study was submitted) is highly supportive of the idea that mutant p53 alters immune signaling and promotes tumourigenesis. The authors should consider adding this reference. (Ghosh M et al, Mutant p53 suppresses innate immune signaling to promote tumorigenesis. *Cancer Cell* 2021 Jan 23;S1535-6108(21)00041-6. doi: 10.1016/j.ccell.2021.01.003.)

Our response:

We have added the following discussion to the manuscript at lines 533-536.

“Ghosh et al. 2021, have reported that mutant p53 reduces the activity of the cytoplasmic DNA sensing cascade which upregulates *IFNB1* to stimulate CD8+, CD4+, and NK cells, while suppressing M2-tumor associated macrophages⁶⁷. However, we found no difference in *IFNB1* expression between tumors bearing mutant, wildtype, or truncated *TP53* (ANOVA p-value: 0.435). Other studies have shown that mutant p53 can interact with *NFκB* to stimulate expression of genes involved in inflammation⁶⁸. Further, interactions within the tumor microenvironment that impact the immune response may exhibit oncogene and tissue specificity⁶⁹⁻⁷¹.”

2. Page 5 line 107: The first quartile of DFI defined here as <123 days and the authors state that tumours from this quartile were enriched for genes in immune-related pathways. But in the paragraph that begins on page 14, line 297 they state the opposite.

Our response:

We have corrected the sentence (see lines 100-101).

“Tumors from the long DFI patients were enriched for genes in immune-related pathways.”

3. Page 36, IHC: There are a few details lacking from these methods that would make it difficult for someone to repeat the experiments: concentration of antibodies, diluent used, incubation and antigen retrieval temperatures, incubation times. These could be placed in a supplemental methods section, but they are important for someone to be able to repeat the findings or compare to their own findings.

Our response:

Please see response from previous comment: “2. IHC and quantification of immune cell infiltrates”.

4. Table 2A and 2B. Some p values are stated as “0” or “0.000”. The stats program may spit out this value, but it should be reported as the actual p value, using an exponent if necessary (as in Table S6C), or being less than a predetermined cutoff, e.g. 0.0001.

Our response:

The actual p-values have now been entered in the tables.

5. Table S1A: The column “In-house sequencing project name” is meaningless to the reader and is confusing. There are patient names (“Bella”, “Rocky”, etc.) that, when paired with all the other data, clearly identify the individual and potentially the owner/client. These should be removed. Likewise, Table S1B includes date of birth, but this not used in the analysis, and is irrelevant and should be removed.

Our response:

We have removed the patient names and the column “In-house sequencing project” from the Table 1A. The date of birth column from Table S1B has been removed.

REVIEWERS' COMMENTS:

Reviewer #2 (Remarks to the Author):

Thanks for your thorough responses to the review and thoughtful revisions. The additional data along with clarification of the methods is very helpful and I'm sure this study will be of great interest to researchers in this field.

(very minor comment: Suppl Fig 8: typo in plot title)

Reviewer #2 (Remarks to the Author):

Thanks for your thorough responses to the review and thoughtful revisions. The additional data along with clarification of the methods is very helpful and I'm sure this study will be of great interest to researchers in this field.

(very minor comment: Suppl Fig 8: typo in plot title)

Response: We have corrected the typo in supplementary figure 8. See page 13 in SupplementaryInformation.pdf file.